# MAPLE (modular automated platform for large-scale experiments), a robot for integrated organism-handling and phenotyping

Tom Alisch[1,2], James D Crall[1,3], Albert B Kao[1], Dave Zucker[4], Benjamin L de Bivort[1,2,4]*

[1]Department of Organismic and Evolutionary Biology, Harvard University, Cambridge, United States; [2]Center for Brain Science, Harvard University, Cambridge, United States; [3]Planetary Health Alliance, Harvard University, Cambridge, United States; [4]FlySorter LLC, Seattle, United States

**Abstract** Lab organisms are valuable in part because of large-scale experiments like screens, but performing such experiments over long time periods by hand is arduous and error-prone. Organism-handling robots could revolutionize large-scale experiments in the way that liquid-handling robots accelerated molecular biology. We developed a modular automated platform for large-scale experiments (MAPLE), an organism-handling robot capable of conducting lab tasks and experiments, and then deployed it to conduct common experiments in *Saccharomyces cerevisiae*, *Caenorhabditis elegans*, *Physarum polycephalum*, *Bombus impatiens*, and *Drosophila melanogaster*. Focusing on fruit flies, we developed a suite of experimental modules that permitted the automated collection of virgin females and execution of an intricate and laborious social behavior experiment. We discovered that (1) pairs of flies exhibit persistent idiosyncrasies in social behavior, which (2) require olfaction and vision, and (3) social interaction network structure is stable over days. These diverse examples demonstrate MAPLE's versatility for automating experimental biology.

DOI: https://doi.org/10.7554/eLife.37166.001

*For correspondence: debivort@oeb.harvard.edu

## Introduction

Genetic model organisms are used to advance our biological understanding in numerous areas including disease and its treatment, basic cell biology, neuroscience and behavior. Species like *Saccharomyces cerevisiae*, *Caenorhabditis elegans,* and *Drosophila melanogaster* are desirable lab model organisms due to their rapid reproduction, ease of rearing, and especially their deep genetic toolkits comprising strains with varying genotypes and transgenic alterations that permit rapid, mechanistic inquiries. To take advantage of these toolkits, screen experiments quantify the phenotypes of hundreds (*Vitaterna et al., 1994*), thousands (*Kain et al., 2012*), tens of thousands (*Ayroles et al., 2015*; *Buchanan et al., 2015*; *Churgin et al., 2017*) or even hundreds of thousands of individual animals (*Robie et al., 2017*). With the ongoing improvement and widespread adoption of high-performance machine vision phenotyping (*Branson et al., 2009*; *Dankert et al., 2009*; *Kabra et al., 2013*; *Kimura et al., 2014*), the time needed to manually handle experimental animals remains the bottleneck limiting data collection.

The advent of liquid-handling robots has radically changed the face of molecular biology, enabling techniques and experiments that had long been imagined, but were too complex, lengthy, or tedious to have been previously realized. But there are no comparable systems for handling of

**eLife digest** Biological research can, at times, be mind-numbingly tedious: scientists often have to do the same experiment over and over on many different samples. When working with animals such as fruit flies, this means researchers have to physically handle large numbers of specimens, selecting certain individuals or moving them from one container to another to perform the study. This represents a serious bottleneck that slows down discovery.

Automation represents an obvious solution to this issue. In fact, it has already revolutionized fields like molecular biology, where robots can handle the liquids required for the experiments. Yet, it is not so easy to automate tasks that involve animals larger than a millimeter. To fill that gap, Alisch et al. have developed a robotic system called Modular Automated Platform for Large-scale Experiments (MAPLE) that can manipulate fruit flies and other small organisms.

Using gentle vacuum, MAPLE can pick up individual flies to move them from one compartment to another. These areas could be places where the insects grow or where experimental measurements are automatically gathered. Putting the robot to work, Alisch et al. used MAPLE to collect virgin female flies for genetic experiments, a common task in fruit flies laboratories. The system was also configured to load flies into arenas where their behavior could be measured. Finally, MAPLE assisted with an experiment that involved tracking the interactions of known individuals to examine if the flies exhibited social networks, and if those networks were stable. This logistically complicated experiment would have been difficult to run without the help of an automated system. Alisch et al. also show that the robot can be adjusted to work with various species often used for research, such as nematode worms, yeast, slime mould and even bumblebees. This allows the system to be useful in a range of research fields.

As MAPLE fits on a table top and is fairly affordable, the hope is that it could help many scientists do their experiments faster and with greater consistency, freeing up time for creative thinking and new ideas. Ultimately, this tool could help to speed up scientific progress.
DOI: https://doi.org/10.7554/eLife.37166.002

experimental organisms that are larger than ~1 mm or cannot be suspended in liquid. There are large-scale systems that handle adult flies in vials, in the form of 'fly flipping' robots. However, these take up a whole room, generally in a core facility, and cost hundreds of thousands of dollars to purchase and maintain, and are therefore inaccessible to most labs. And while there are many examples of high-throughput phenotypic assays in *Drosophila* (*Branson et al., 2009*; *Kabra et al., 2013*; *Kain et al., 2012*; *Buchanan et al., 2015*; *Geissmann et al., 2017*), the systems that are not single-purpose still require human intervention to load and unload individual flies. Some researchers have used flies' natural tendency to climb up (negative gravitaxis) to isolate individuals for behavioral analysis (*von Reyn et al., 2014*), imaging (*Medici et al., 2015*), or microsurgery (*Savall et al., 2015*). But dependence on this particular behavior fundamentally caps throughput, inadvertently selects for a subset of a population, and limits eligible genotypes. An alternative approach, actively conveying flies with airflow (*MacMillan and Hughson, 2014*) permits moving animals on demand, and opens the door for increased throughput. The dearth of instruments for automating *Drosophila* experiments is representative of the situation for many other lab organisms, such as yeast and *C. elegans*, where there is no standard platform for automated handling of the organisms themselves.

Here, we present an automated platform that is high-throughput and flexible enough to assist in conducting diverse experimental protocols in *Drosophila* and other species. Due to its modular design, the system can automate diverse assays in a wide variety of organisms (including yeast, *C. elegans*, the slime mold *Physarum polycephalum*, and the bumblebee *Bombus impatiens*, an assortment chosen to demonstrate the platform's versatility). For fruit flies, where we developed significant capabilities, this instrument can conduct numerous protocols, including loading of individual fruit flies for circadian rhythm (*Pfeiffenberger et al., 2010*; *Geissmann et al., 2017*) experiments, or aiding with lab chores like collecting virgin female flies for genetic crosses or passaging individual flies in controlled culture conditions for longevity assays. The physical platform of this instrument integrates animal husbandry and phenotyping, permitting end-to-end experimental protocols. Its low cost (~$3,500), programmability, and scalability permit large-scale experiments that take

advantage of the many benefits lab model organisms offer, such as huge fly genetic libraries containing thousands of lines (*Jenett et al., 2012*; *Thibault et al., 2004*). After demonstrating MAPLE's breadth of utility across species, we highlight the depth of its capabilities with two particularly time- and manual labor-intensive fruit fly tasks: rapidly collecting virgin females, and large-scale longitudinal measurement of fly social networks and behavior. The latter experiment reveals previously unknown stability of *Drosophila* social interactions, and confirms that both olfaction and vision are required for dyad-specific social interactions in this context.

## Results

### MAPLE physical implementation

With the high level goals of modularity, scalability, and automatability in mind, we designed the MAPLE system with the following design constraints: (1) It features a large, flat experimental workspace with room for multiple flexibly-configurable experimental modules. (2) This workspace is physically open for user convenience, and transparent on the top and bottom for in situ optical phenotyping. (3) Multiple end-effectors can move throughout the workspace to handle organisms, capture images, and manipulate experimental modules. (4) It features failsafe mechanisms so that users can leave it unattended without worrying that it would damage itself or experimental modules. (5) It is relatively inexpensive and scalable.

MAPLE (*Figure 1*, *Figure 1—figure supplement 1A*) was built using extruded aluminum rails to support x-, y-, and z-carriages mounted on linear rails in a Cartesian configuration. We employed the CoreXY system (*Moyer, 2012*), which reduces the mass of the moving part of the X/Y gantry by fixing the stepper motors on the frame. (However, for the speeds at which we run MAPLE, which are roughly 80% as fast as human hands conducting experiments (*Video 1*), mounting the y-axis stepper motor on the x-axis carriage would likely not reduce performance.)

With respect to our design constraints: (1) the accessible experimental workspace measures 100 × 28.2×7.5 cm on the x-, y-, and z-axes, respectively. Its floor is clear acrylic with cable/tubing pass-throughs. Locating brackets (laser cut out of 6 mm acrylic) were affixed to an interchangeable acrylic surface with the same footprint as the floor (a 'workspace plate'), allowing experimental modules to be precisely and repeatably positioned within and removed from the workspace (*Figure 1—figure supplement 2*). Interchanging workspace plates allows rapid reconfiguration of the workspace for different experimental procedures. (2) The sides, top, and bottom of MAPLE are open or made of clear acrylic, permitting the optical phenotyping of flies in experimental modules at all time points other than when the end-effector carriages are above the modules. (3) The end-effector assembly comprises three independent z-axes, each featuring a single tool (*Figure 1A*; *Video 2*): an object manipulator for picking up experimental module components like plastic lids using vacuum; a USB digital camera with LED illumination for acquiring high-resolution images for machine-vision; and an organism manipulator for handling small individual animals using vacuum, or, in the case of our yeast experiments, wooden applicators (*Figure 1—figure supplement 3*). (4) All motion axes have physical limit switches and/or software limits preventing overtravel. The organism manipulator end effector, which is rigid and must align precisely with experimental modules at different heights, is equipped with a collision-detection switch to halt z-motion before the robot is damaged. This sensor can also be used to detect the height of rigid module components. It is safe to leave MAPLE unattended (*Video 3*). (5) MAPLE components cost approximately $3500. Its bill of materials (*Supplementary file 1*, assembly instructions (*Supplementary file 2*), and code libraries (see Materials and methods for links) have been made public under open source licenses. MAPLE measures 43.5 cm in the y-dimension permitting mounting in standardized 19' rack systems, so multiple robots can be arranged compactly.

### MAPLE conducts experiments on numerous species

We first established that MAPLE can be used to automate experiments in a wide variety of lab organisms. Specifically, we implemented experimental MAPLE protocols for baker's yeast *S. cerevisiae*, the nematode *C. elegans*, the slime mold *P. polycephalum*, and the fruit fly *D. melanogaster* (*Figure 1B–E*). For yeast, we programmed MAPLE to transfer yeast cells from a single colony on a source plate to target plates and streak the cells out to grow new colonies from single cells

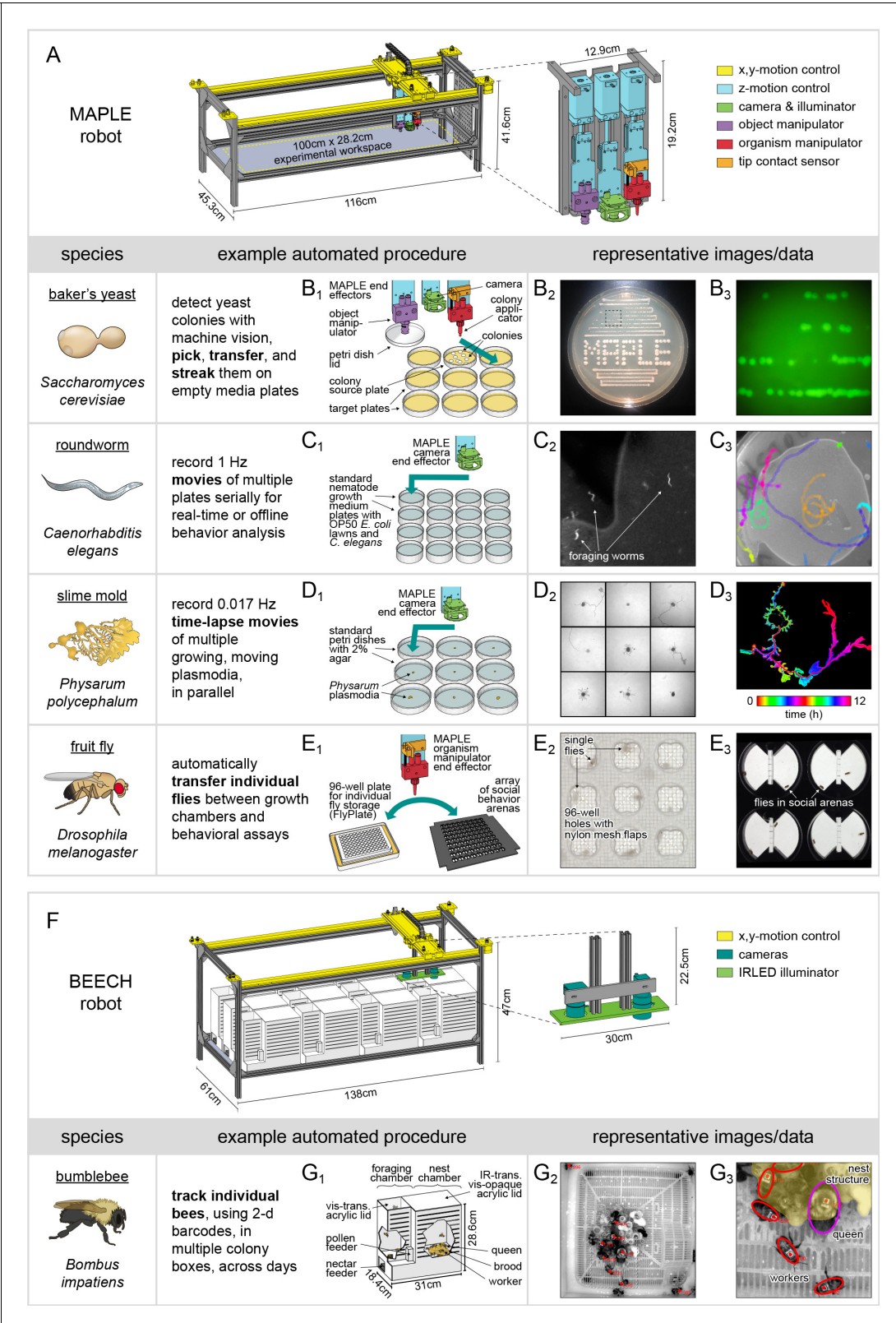

**Figure 1.** MAPLE, a robotic platform for conducting experiments with many species. (**A**) Schematic of MAPLE with workspace dimensions indicated and an expanded schematic of the end effectors. Colors indicate different robotic systems. (**B–F**) Demonstrations of MAPLE experiments in yeast, *C. elegans*, *Physarum polycephalum* and fruit flies. (**B1**) Yeast colonies were automatically picked and transferred to fresh media plates in a pattern (**B2**), and then streaked to grow colonies from single cells (B2, **B3**). B3 is an image of GFP fluorescence in the colonies demarcated by the inset box of B2. *Figure 1 continued on next page*

*Figure 1 continued*

(**C1**) Worm locomotion on standard culture plates with OP50 bacterial lawns was captured at 1 Hz using MAPLE's camera effector. (**C2**) Cropped view of one such image showing individual worms. (**C3**) Tracks of multiple worm motion across a plate over 8.5 min, showing the field of view MAPLE's camera and potential for motion phenotyping. (**D1**) Time-lapse movies of the movement of *Physarum* plasmodia were recorded in parallel for nine cultures at imaging rates of 0.017 Hz (one image/min) for 12 hr. (**D2**) Composite of plasmodia morphologies approximately 9.6 hr into the 12 hr recording. (**D3**) Time-coded image of the growth of a single plasmodium, overlaying images of the plasmodium over time in colors cycling over hues. (**E1**) Representative MAPLE procedure for moving fruit flies between a growth/storage module (left) and a phenotyping module (right) using the organism manipulator. (**E2**) Photo of flies in a FlyPlate single-housing storage module. See description below. (**E3**) Photo of flies in a social arena phenotyping module. See description below. (**F**) Bee Experimental Ethology Colony Hardware (BEECH): a MAPLE-derived robotic platform for imaging bumblebee behavior. BEECH was used to record bumblebee behavior in multiple colonies over long periods of time (up to 7 days). The end effectors of this robot were IR-sensitive cameras (right) for recording digital video of bees behaving. Bees were housed in acrylic colony boxes (**G1**) with a dark nest compartment and circadian-lit foraging compartment where they, respectively, reared developing young and collected nectar and pollen from feeders. (**G2**) Image acquired from the BEECH IR camera showing automatically tracked bees in the nest chamber. Inset box expanded in (**G3**) to illustrate workers (red ellipses), the queen (magenta ellipse), and nest structure (yellow overlay), all of which are tracked in BEECH data sets. This figure is also a target for an augmented reality view of MAPLE. Print the figure in portrait orientation on letter paper, place the printout on a horizontal surface like a bench or desk, and view it in scan mode in the 'Augment' mobile app to interact with a to-scale rendering of MAPLE.

DOI: https://doi.org/10.7554/eLife.37166.003

The following figure supplements are available for figure 1:

**Figure supplement 1.** MAPLE hardware architecture.

DOI: https://doi.org/10.7554/eLife.37166.004

**Figure supplement 2.** MAPLE experimental workspace.

DOI: https://doi.org/10.7554/eLife.37166.005

**Figure supplement 3.** Details of the fly vacuum tip for the organism manipulator.

DOI: https://doi.org/10.7554/eLife.37166.006

(*Figure 1B*, *Video 4*). For *C. elegans*, we used MAPLE's camera effector to record 1 Hz movies of worms foraging on a lawn of OP50 bacteria (normal *C. elegans* culture conditions; *Figure 1C*). We tracked these worms offline to produce *Video 5*, demonstrating that MAPLE's camera is of high enough fidelity to capture worm behavior on this spatial scale. Collecting such movies from multiple plates serially would allow MAPLE to conduct behavioral screens. Next, for the slime mold *Physarum*, we plated nine plasmodia on nine plates of 2% agar (*Figure 1D*). We programmed MAPLE to take a photo of each plate once per minute and recorded plasmodia movement over the next 12 hr (*Video 6*). From these images, we compiled a combined time-lapse (*Video 7*) of plasmodial outgrowth and motion, which contrasts the exploratory behaviors of the different individuals. Lastly, for fruit flies, we developed a variety of experimental modules that can be flexibly reconfigured to conduct numerous experiments. These are detailed in the rest of the paper, along with new scientific results obtained with them. Automated fruit fly experiments generally exploit MAPLE's organism manipulator to move individual flies between compartments where the flies are grown or housed and phenotyping compartments where experimental data are automatically collected (*Figure 1E*).

MAPLE's modularity, open-source design, and hierarchical software architecture also facilitates hardware modifications that expand its multi-species capabilities. We adapted the basic MAPLE design for high-throughput imaging of uniquely identified workers within colonies of the Common Eastern bumblebee (*Bombus impatiens*). In the Bee Experimental Ethology Colony Hardware (BEECH) system (*Figure 1F,G*), up to 12 colonies of ~50 bumblebees each are housed in acrylic colony boxes featuring a dark nest chamber (*Video 8*) and a circadian-lit foraging chamber, where bees are supplied with nectar and pollen. This MAPLE-derived two-dimensional Cartesian robot moves a multi-camera

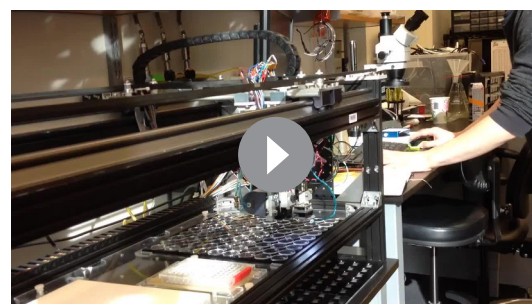

**Video 1.** MAPLE in use. MAPLE is situated on a standard experimental bench. The user controls its behavior through an attached PC. Real-time video shows that MAPLE moves at speeds similar to human fly experimentalists.

DOI: https://doi.org/10.7554/eLife.37166.007

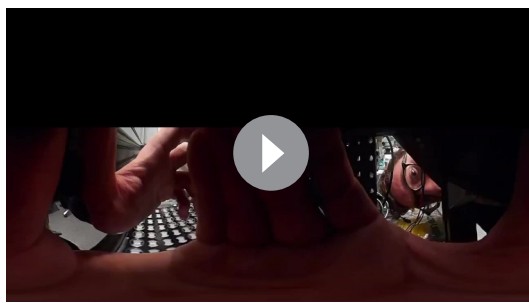

**Video 2.** Motion VR view from within MAPLE during an experimental procedure. Time-lapse view from a wide-angle camera mounted on the x-axis assembly of MAPLE during a procedure to move flies from a 96-well plate into social arenas, providing a clear view of the action of all three z-assemblies and end effectors. Moments when MAPLE flashes multiple times successively in the same position reflect an algorithm to vary the exposure time for image acquisition to detect the opening of a multi-position loading port. This video is available on YouTube at https://youtu.be/wxEgbYfif_M. The viewing angle can be adjusted during playback in the YouTube viewer.
DOI: https://doi.org/10.7554/eLife.37166.008

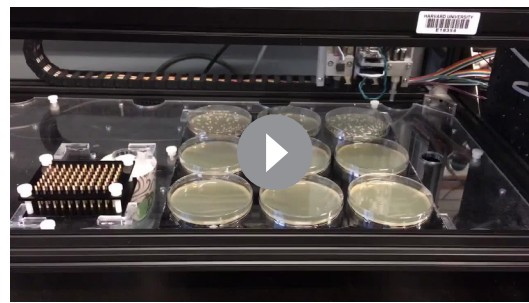

**Video 3.** MAPLE can operate autonomously. Wide-angle movie acquired by mounting the camera on the y-assembly as MAPLE conducts a social arena array loading procedure. After starting the protocol, confirming that the camera stays attached and trimming some zip-tie tails from the camera mount, the operator walks away while MAPLE continues the experiment.
DOI: https://doi.org/10.7554/eLife.37166.009

end-effector from colony to colony, recording high-resolution video of the nest and foraging chambers (*Videos 8* and *9*), which permits automated tracking of individual tagged bees using machine vision (*Crall et al., 2015*) for up to several weeks. While replacing MAPLE's z-axis assembly with multiple cameras and extending all three dimensions of MAPLE's frame to accommodate the 12-colony box modules, BEECH employs identical construction and motion-control, demonstrating the versatility of the MAPLE design.

## MAPLE-handled flies exhibit normal behavior

An important step in adopting a new automated approach is confirming that the procedure generates similar data to prior manual experiments. To do this, we focused on fruit flies, and set out to confirm that handling by MAPLE did not damage flies or introduce discrepancies compared to experiments conducted manually. Specifically, we examined the locomotor performance of flies handled by MAPLE and unhandled flies. First, we manually aspirated 96 anesthetized flies into a FlyPlate (a modified 96-well plate for single fly storage. Detailed description below). After 2 hr of rest, half the flies (48) were subjected to repeated manual removal and replacement back into their well using the MAPLE organism manipulator, while the remaining flies (48) were left unhandled (*Figure 2A,B*). After this handling procedure, the entire plate was imaged in a backlit motion tracking rig (*Buchanan et al., 2015*). Handling and imaging were then repeated an additional four times to test for cumulative effects. Flies handled by MAPLE

**Video 4.** Transferring yeast colonies with MAPLE. Real-time video showing MAPLE load an applicator stick in its organism manipulator, pick a colony from a source plate, transfer those cells to a target plate in a known pattern, repeat this procedure for additional single colonies from the source plate, and finish by streaking out the last transferred colony.
DOI: https://doi.org/10.7554/eLife.37166.010

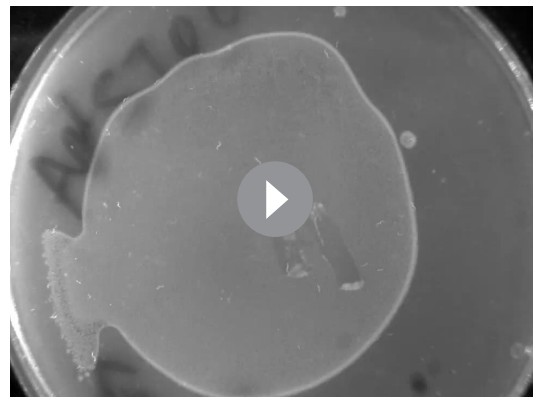

**Video 5.** C. elegans worms foraging on growth plates. MAPLE collected 1 Hz video at 1944 × 2592 px resolution of worms foraging on a standard growth plate with a lawn of OP50 *E. coli*. Worm positions were recorded manually in Fiji, and the track overlays were added in MATLAB. Movie frames acquired at 1 Hz and played back at 30 fps.
DOI: https://doi.org/10.7554/eLife.37166.011

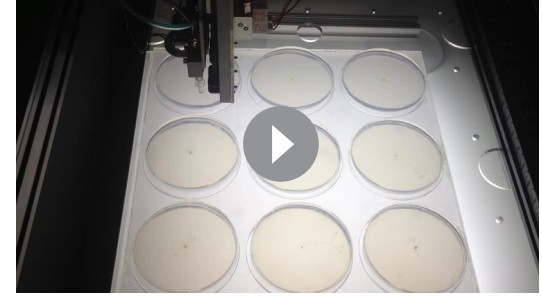

**Video 6.** MAPLE collecting images for time-lapse movies of Physarum growth. Real-time video of MAPLE traveling between plates of 2% agar inoculated with *Physarum* plasmodia, and taking an image of each plate at a rate of 1 image/min. MAPLE thus collected time-lapse movies (*Video 7*) of all the plasmodia in parallel.
DOI: https://doi.org/10.7554/eLife.37166.012

were statistically indistinguishable from unhandled flies both in the fraction that were active across imaging sessions (*Figure 2C*; $b = -0.02$, $t(477) = -0.098$, p=0.92 by multinomial logistic regression) and in their mean walking speed across imaging sessions (*Figure 2D*; F(4, 376)=0.51, p=0.73 by mixed-effects ANOVA).

Second, we measured the locomotor bias of individual flies in Y-shaped mazes (*Ayroles et al., 2015*; *Buchanan et al., 2015*) configured with multi-position loading ports. Awake flies were loaded into these mazes from FlyPlates by either manual aspiration or MAPLE-handling (*Figure 2E,F*). The across-individual distribution of walking speeds was statistically indistinguishable between the manual and MAPLE-handled group (*Figure 2G*; p=0.61 by KS-test). Likewise, the across-individual distribution of turning bias (the tendency of individuals to turn left or right at the choice point in the center of the Y-maze) was indistinguishable between handling treatments (*Figure 2H*; p=0.74 by KS-test). The same was true for comparisons of manual and MAPLE-handled behavior in semi-circular arenas (*Figure 2— figure supplement 1*).

## An ecosystem of MAPLE experimental modules and software

Having confirmed that MAPLE does not obviously distort fly behavioral data, we set out to create an ecosystem of reconfigurable modules that could be used combinatorially in the MAPLE workspace to conduct a large number of experimental protocols. We fabricated and deployed a number of modules, which fall into three categories (*Figure 3*): fly source, fly sink, and phenotyping modules. Fly source modules are repositories from which flies can be removed in a controlled fashion and transferred into downstream modules. These include the Fly

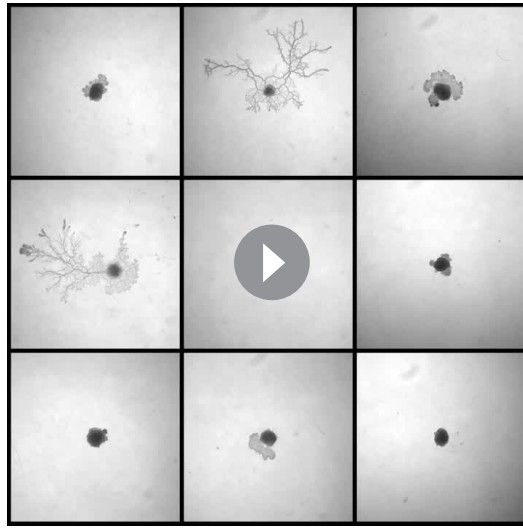

**Video 7.** Time-lapse movies of Physarum plasmodial growth and movement . Movie frames acquired at 0.017 Hz and played back at 30fps, covering a recording period of 12 hr. Movies show outgrowth followed by exploratory motion growth as well as rhythmic cytosolic pumping.
DOI: https://doi.org/10.7554/eLife.37166.013

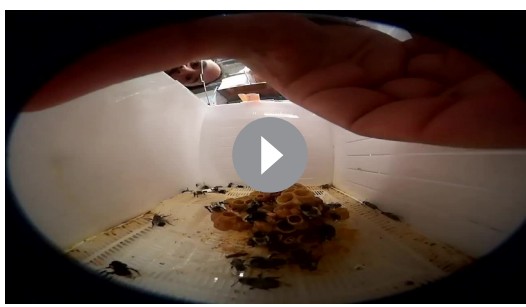

**Video 8.** Motion VR view from within a bumblebee colony in BEECH. Wide-angle movie acquired by mounting the camera inside a bumblebee colony box and placing into the BEECH platform for imaging. BEEtag spatial barcodes are visible on the backs of individual bees. Once BEECH begins to move, its camera and IR illuminator effectors periodically come into view above the nest chamber (e.g. at 3:05); in this movie, the normally visible-opaque next chamber roof has been replaced with clear acrylic. This video is available on YouTube at https://youtu.be/crIb4ZfecYQ. The viewing angle can be adjusted during playback in the YouTube viewer.
DOI: https://doi.org/10.7554/eLife.37166.014

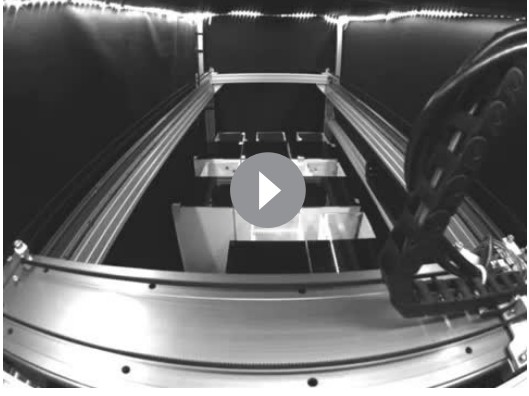

**Video 9.** BEECH recording behavior of multiple bumblebee colonies. Timelapse of a real BEECH experiment in which the camera end effectors are moved between bumble bee colony boxes to record brief videos in each successively.
DOI: https://doi.org/10.7554/eLife.37166.015

Dispenser (FlySorter LLC), a small device that outputs single flies, on demand, from a standard plastic vial pre-loaded with many flies. It can be triggered to dispense a fly via a serial command sent over USB. A standard $CO_2$ pad with a porous polyethylene surface, used to anesthetize flies manually at the start of a MAPLE session, serves as a source of flies. Using machine vision, MAPLE is capable of recognizing flies' positions on the pad. FlyPlates (FlySorter LLC) are modified 96-well plates in which the floor has been replaced with a metal mesh, allowing flies stored in the wells to feed on fly media below the plate. The lid features a nylon mesh with X-shaped slits cut above each well (*Figure 2B*), which allow an aspirator tip or the organism manipulator end-effector to enter the well, retrieve or deposit a fly, and leave the well without permitting the fly to escape. Because flies can be deposited in this module, it also falls in the category of fly sink modules.

Fly sink modules are destinations into which flies that have been handled by MAPLE can be deposited. In the case of the FlyPlate, deposited flies can be later removed. Other fly sink modules are one-way, including a morgue, a dish of slightly soapy water or 70% ethanol covered with a nylon mesh lid in the style of the FlyPlate. Standard fly culture media vials with nylon mesh loading adapters can be used to collect many flies after MAPLE handling for long-term storage.

The last category of modules is phenotyping modules, which produce experimental data. We have created a number of behavior phenotyping modules (*Figure 3—figure supplement 1*), including arrays of circular open field arenas, arrays of Y-shaped mazes for measuring locomotor handedness (*Buchanan et al., 2015*), and social arena arrays, in which pairwise social interactions can be monitored. Phenotyping modules are not limited to collecting behavioral data. For example, flies can be loaded into standard 96-well plates outfitted with nylon mesh lids. From there, they are ready for molecular protocols, including use by liquid-handling robots. Flies in FlyPlates, which have access to food, can be used for circadian assays (*Tataroglu and Emery, 2014*), longevity (*Stearns et al., 2000*), or pharmacological experiments (*Gasque et al., 2013*). Modules such as the Fly Dispenser can be situated outside MAPLE's frame, receiving and delivering flies via air-flow tubing. 3D-printed adapter blocks connect these tubes into the MAPLE workspace and are situated with locating brackets.

The modularity of MAPLE's hardware is reflected in its software as well (*Figure 3—figure supplement 2*). Each experimental procedure is associated with a Python experimental script file. These files call functions that (1) implement common multi-step robot actions (like retrieving a fly from a behavioral arena), (2) implement low-level elemental robot actions, (3) are specifically associated with the modules used in a particular experiment, and (4) mediate the remote control of MAPLE

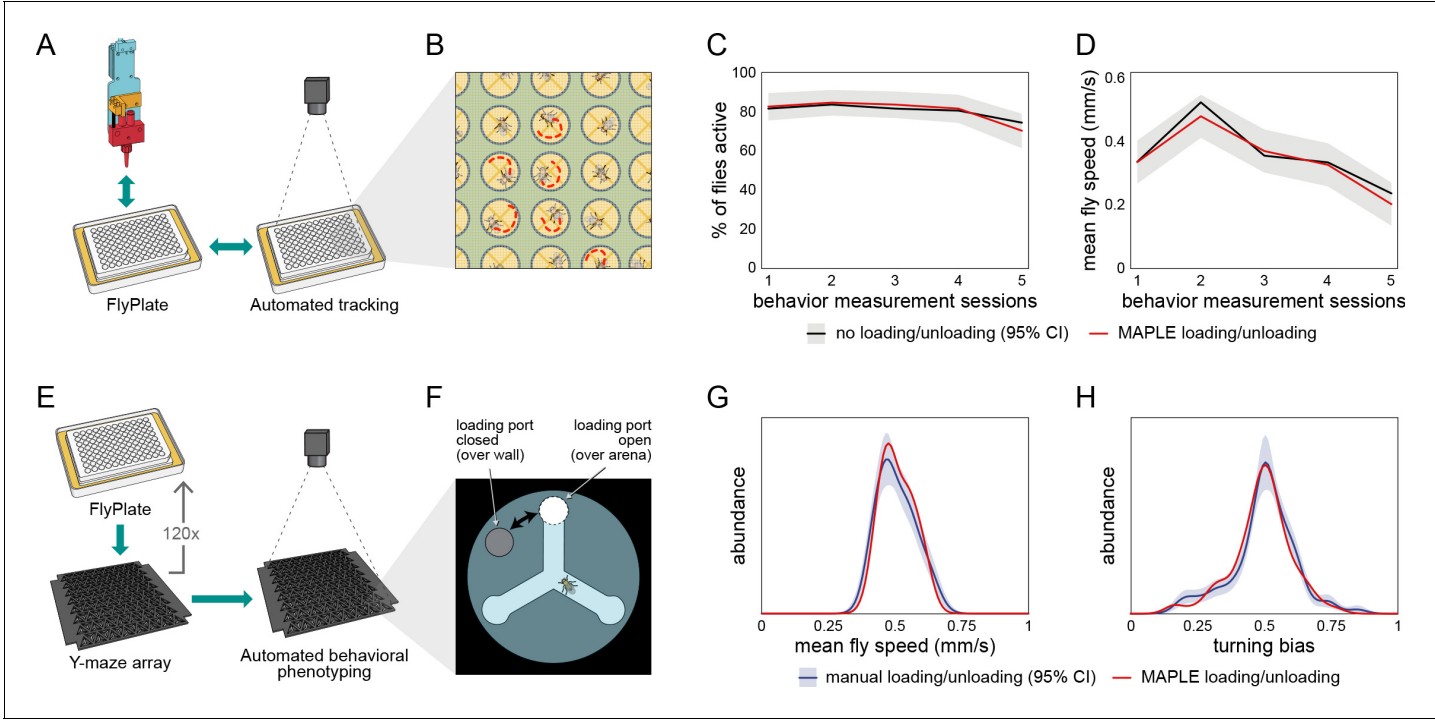

**Figure 2.** Behavior of flies manipulated by MAPLE is similar to manual controls. (**A**) Diagram of repeated MAPLE handling procedure. Flies were removed from and replaced back into wells of a FlyPlate using MAPLE's organism manipulator. In between these handling events their activity was assessed by automated tracking. (**B**) Illustration of automatic tracking of flies in a FlyPlate through the nylon mesh covering the wells. (**C**) Percentage of flies exhibiting supra-threshold activity (mean speed >0.1 mm/s) across handling sessions. Red line is MAPLE-handled flies, black line is matched unhandled flies. Gray area denotes 95% CI around unhandled flies. (**D**) Fly mean speed across handling sessions. (**E**) Illustration of MAPLE-assisted handedness phenotyping procedure. (**F**) Y-maze arena adapted for MAPLE use with a multi-position loading port allowing the deposition of awake flies into the behavioral arena. The hole in the rotatable lid can be aligned to the arena to load or unload flies, or to the wall surrounding the arena to trap flies in. (**G**) Distribution of fly mean speeds in MAPLE and manually loaded experiments. Distributions shown are kernel density estimates (KDEs). Gray area is 95% CI around manually loaded flies as estimated by bootstrap resampling of KDEs. (**H**) Distributions of fly turning bias (# right turns / # total turns) in MAPLE and manually loaded experiments.

DOI: https://doi.org/10.7554/eLife.37166.016

The following figure supplement is available for figure 2:

**Figure supplement 1.** MAPLE- versus manual-handling in the social arena.
DOI: https://doi.org/10.7554/eLife.37166.017

over the internet. This software architecture permits the rapid scripting of new experimental protocols at a high level (*Figure 3—figure supplement 3*).

To demonstrate the experimental flexibility of this ecosystem of modules and software, we implemented a number of experimental procedures (*Figure 3B*). These include: (1) collecting virgin flies for genetic crosses by dispensing flies as they eclose and then distributing them into individual wells of FlyPlates where their isolation preserves their virgin status indefinitely; (2) loading flies into Y-shaped arenas to measure their locomotor biases, the time-consuming step of a routine assay in our lab; (3) loading flies from the Fly Dispenser into the FlyPlate wells for long-term culturing and circadian phenotyping; and (4) loading flies into social arenas to measure their pairwise interactions. Below we describe results from procedures (1) and (4) in detail, as well as the scientific findings obtained using the latter procedure.

## MAPLE-assisted virgin picking is more efficient than manual methods

We tasked MAPLE with performing a tedious task that consumes great amounts of time in essentially all *Drosophila* labs: collecting virgin females for genetic crosses. Female *D. melanogaster* will not mate with males for approximately 6 hr after they eclose from the pupal case. In the first portion of this interval, they have morphological characteristics (puffy abdomens, translucent cuticle, and visible

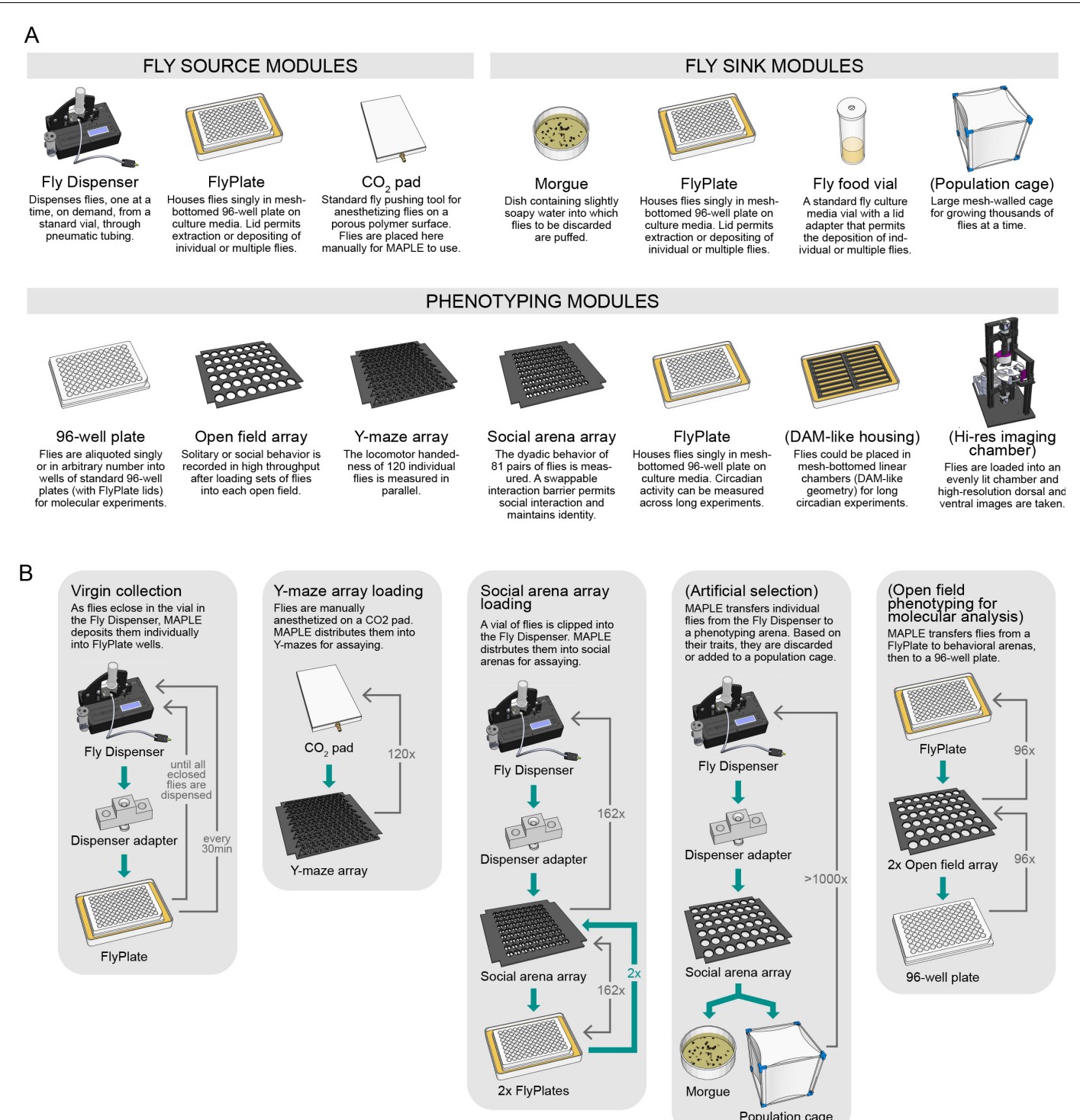

**Figure 3.** An ecosystem of MAPLE hardware modules for flexible experimental automation in fruit flies. (**A**) Illustrations and brief descriptions of experimental MAPLE modules. Fly source modules can provide flies into the MAPLE system. Fly sink modules can receive flies. Phenotyping modules are used to collect experimental measurements. See Materials and methods for more extensive descriptions. Modules with names in (parentheses) are under development. (**B**) Simplified flowcharts illustrating a selection of tasks MAPLE is capable of performing using different combinations of modules. Green arrows indicate the flow of animals through the task, thin grey arrows the motion of the MAPLE end effectors. Tasks with names in (parentheses) are hypothetical and illustrate the scope of possibilities.

DOI: https://doi.org/10.7554/eLife.37166.018

*Figure 3 continued on next page*

*Figure 3 continued*

The following figure supplements are available for figure 3:

**Figure supplement 1.** Multi-position loading ports.
DOI: https://doi.org/10.7554/eLife.37166.019
**Figure supplement 2.** MAPLE software architecture.
DOI: https://doi.org/10.7554/eLife.37166.020
**Figure supplement 3.** MAPLE procedure flowcharts.
DOI: https://doi.org/10.7554/eLife.37166.021

meconium in the gut) that correlate with their young age and are reliable, but conservative, indicators of virginity. In traditional manual virgin-picking, only females with these morphological correlates are collected. This means that many virgin females lacking morphological correlates in the latter portions of the 6 hr no-mating window are discarded, unless practitioners collect virgins from a stock vial or bottle at regular intervals at least three or four times a day.

MAPLE has the potential to recover 100% of females as virgins by isolating them quickly after they eclose, and storing them individually to preclude mating. To implement this procedure, we devised a simple custom fly media vial in which the lower portion containing fly food is detachable. Parental generation flies lay eggs in this food, or food from another vial containing larval flies can be transferred into the custom vial. When flies in this experimental generation climb onto the walls of the vial and pupate, the food-containing portion is manually detached, and replaced with an empty vial-bottom (*Video 10*). This pupae-containing, foodless vial is then placed in the Fly Dispenser. Every 30 min, all newly eclosed flies are dispensed into MAPLE, which distributes them individually into the wells of a FlyPlate (*Figure 4A*).

This process was not without error: at rates of ~3% a well was left empty, or ~8% a well was loaded with two flies. A representative outcome is shown in *Figure 4—figure supplement 1*. To recover only the virgin females, fully loaded FlyPlates were removed from their media base, brought to traditional $CO_2$ fly-pushing pads at dissecting microscopes, the flies were anesthetized through the wire mesh floor, and then the plate and pad assembly was inverted, leaving the flies on the $CO_2$ pad in their respective positions from the wells. The sex of each fly was determined by eye, and females that had been stored either alone, or with no males were collected. In a head-to-head comparison of typical manual virgin-collecting (picking at the beginning and end of the work day from five vials) versus MAPLE virgin-collecting, virgin females were procured at a rate of 1.5/min (including the time needed to bring vials from the incubator, anesthetization, etc.) and 4.9/min using MAPLE (including the time to set up the pupa vial, load the dispenser, manually sex and sort the flies, etc.) (*Figure 4B*). In the future, we anticipate MAPLE will be able to sex flies without human intervention using the high-resolution imaging module, further decreasing manual work required.

## MAPLE reveals persistent fly social interaction networks and their sensory basis

Lastly, we set out to see if MAPLE could conduct experiments that would be difficult using traditional manual methods. Specifically, we set out to measure social interaction networks (SINs; *Schneider et al., 2012*; *Pasquaretta et al., 2016*) between pairs of flies, and then determine if measures of pair-wise affiliation are preserved on the timescale of days (*Figure 5*). It is known from group behavioral experiments that individuals spend more time interacting with specific other individuals (*Simon et al., 2012*), and such interaction-based SINs remain stable over tens of minutes in fruit flies (*Schneider et al., 2012*). In other species, such as the forked fungus

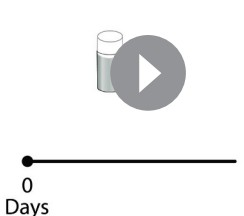

0
Days

**Video 10.** MAPLE-assisted virgin-collecting procedure. Animation illustrating the MAPLE procedure for efficiently collecting virgin female flies. See also *Figure 4*.
DOI: https://doi.org/10.7554/eLife.37166.024

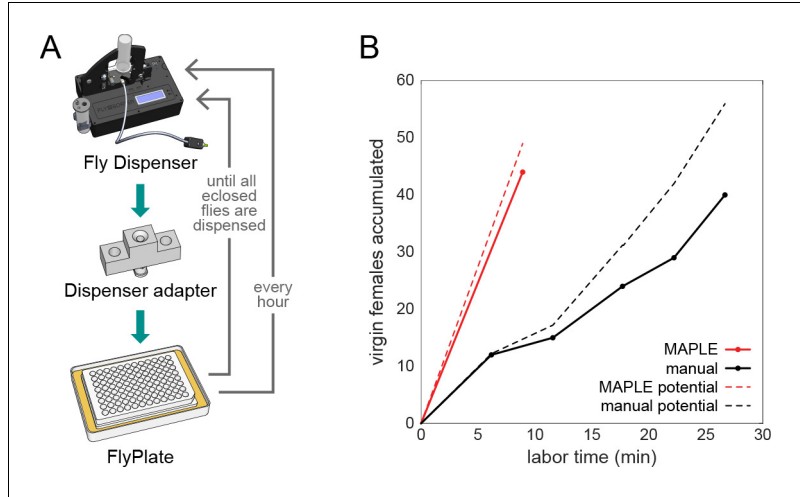

**Figure 4.** MAPLE performance at collecting virgin female flies for genetic crosses. (**A**) Diagram of modules and procedure employed in MAPLE virgin-collecting. (**B**) Comparison of cumulative number of virgin females collected versus manual labor time required in MAPLE and conventional manual virgining procedures. Dotted lines indicate the maximum potential virgins that might have been collected in each approach. Thus the difference between the dotted and solid line indicates females that were 'lost' and not retained as virgins.
DOI: https://doi.org/10.7554/eLife.37166.022
The following figure supplement is available for figure 4:

**Figure supplement 1.** Outcome of a virgin-collecting experiment.
DOI: https://doi.org/10.7554/eLife.37166.023

beetle *Bolitotherus cornutus*, these dyadic interactions are stable for days (*Formica et al., 2017*). Because it is challenging to perfectly maintain individual identity through experiments in which multiple individuals are in the same compartment and subsequently retrieved, stored and retested, it is unknown if dyad-specific affiliative measures are stable over longer periods of time in flies. To assess this we devised a new high-throughput assay to measure affiliative behavior between pairs of flies (*Figure 5A,B*). This consisted of a 9 × 9 array of adjacent, approximately semi-circular arenas separated by an interchangeable barrier. MAPLE can load individual flies into each arena-half using a multi-position loading port (*Video 11*). We made four versions of the interchangeable barrier: an open-clear barrier was made of clear acrylic with grooves to connect the half-arenas, permitting flies to see and smell each other (*Video 12*); a solid-clear barrier permits flies to see each other but impedes airflow; an open-black barrier permits airflow but blocks visual cues; and a solid-black barrier blocks both airflow and visual cues. Flies in this assay are thus presented with the choice of interacting with a designated partner across the barrier, or not.

As a measure of social affiliation, we determined an 'interactivity index,' defined as the Pearson correlation coefficient between the distances of each fly to the barrier over time (*Figure 5C,D*). Either significantly positive or negative values of this index indicate social interaction. We found that male-female and female-female, although not male-male, dyads produced the same distribution of interactivity indices (*Figure 5—figure supplement 1A*). Most social experiments were performed with virgin female-female dyads, although for some control groups we also included male-female dyads.

We programmed MAPLE to load pairs of flies from the FlyPlate into social arenas according to three different dyad schemes (*Figure 5E*). These varied from high-throughput (testing 162 flies at a time), with each fly tested in just a single dyad, to low-throughput (10 flies total) but saturated with respect to all possible dyadic pairings. Using the high-throughput scheme, we first measured the interactivity indices of wild type (Canton-S) flies separated by open-clear barriers. The observed distribution was significantly different compared to shuffled controls that scramble dyadic pairings (*Figure 5—figure supplement 1B*; p<0.0001 by KS-test), indicating that interactions are dyad-specific.

Dyads separated by barriers that limit sensory cues exhibited significantly lower mean absolute interactivity indices (*Figure 5F*), indicating that both olfactory and especially visual cues drive

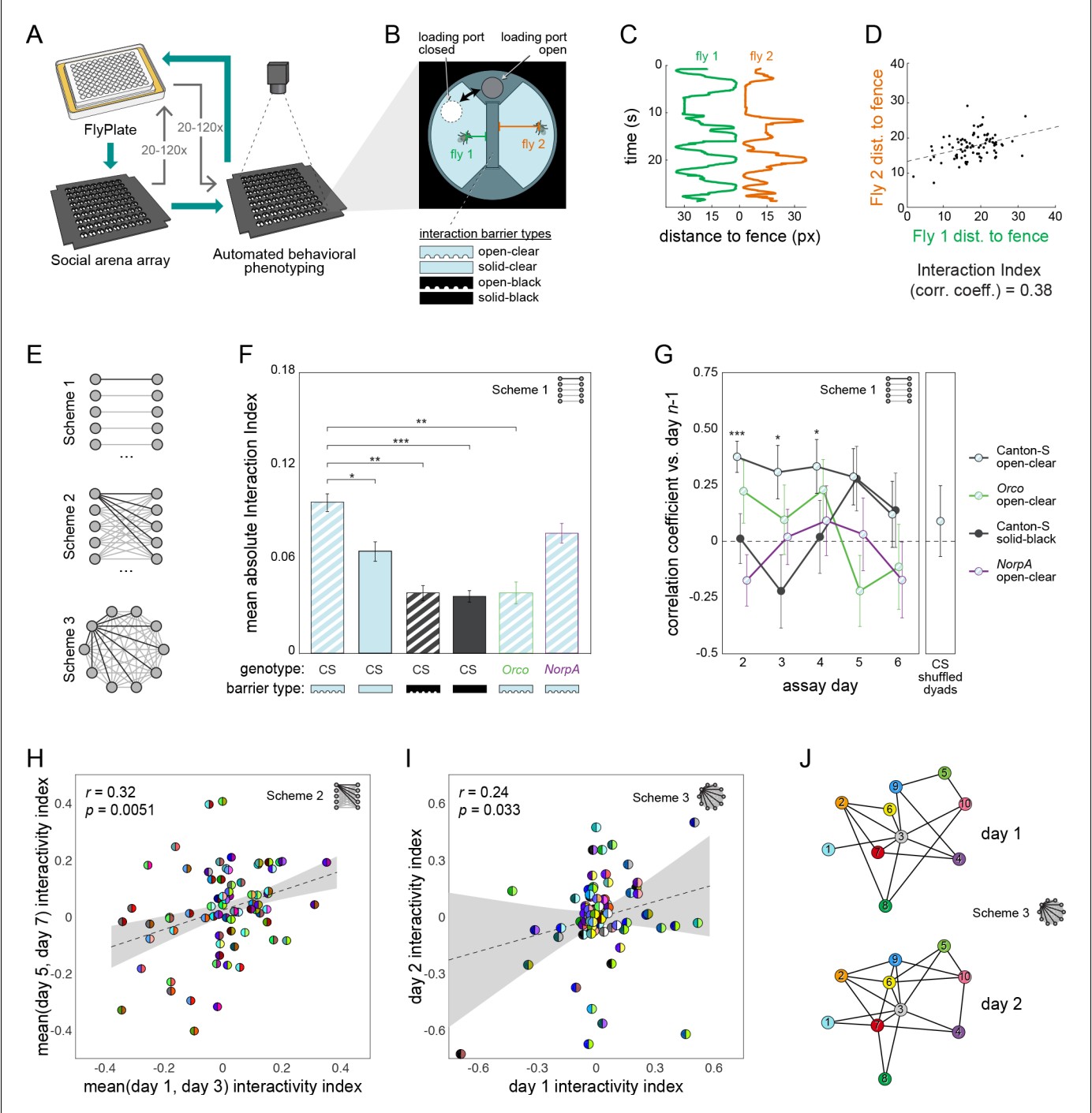

**Figure 5.** Persistent fly social networks measured using MAPLE. (**A**) Diagram of MAPLE-assisted social interaction behavior phenotyping procedure. (**B**) Schematic of one of 81 social arenas including interchangeable interaction barrier types (shown in side-on view to illustrate the channels along the bottom of some barriers that permit airflow). A multi-position loading port allows flies to be loaded or removed from either compartment independently. (**C**) Individual flies' distances to the interaction barrier over time in an example session. Colors correspond to distances illustrated in (**B**). (**D**) An example dyad interactivity index, calculated as the correlation between individual flies' interaction barrier distances. (**E**) Diagram of pairing schemes employed to form dyads. (**F**) Mean absolute interactivity indices across interaction barrier types and genotypes. Dyads were formed according to scheme 1. Patterns denote barrier type, outline colors denote genotype. Error bars correspond to ±1 SEM. Mean absolute interactivity indices differ significantly across conditions, $F_{(7, 853)}=13.42$, $p<0.001$. Asterisks indicate pair-wise comparisons that are significant by t-test. *: significance at $\alpha = 0.05$; **: significance at $\alpha = 0.01$; ***: significance at $\alpha = 0.001$. (**G**) Pearson correlation coefficient between dyad interactivity indices measured on successive days, over 6 days. Dyads were formed according to scheme 1. Point patterns denote barrier type, line colors denote genotype. Error bars

*Figure 5 continued on next page*

*Figure 5 continued*
correspond to ±1 SEM. Asterisks denote significant two-tailed z-tests. (**H**) Scatter plot of interactivity indices of Canton-S (wild type) flies in arenas with open-clear barriers from measurements made across 2- to 4-day intervals. Dyads were formed according to scheme 2. Gray area is 95% CI of the linear regression line. Circles represent dyads; semi-circles denote individual flies forming a dyad; colors denote fly identity. Pearson correlation coefficient is statistically significant, $r(99) = 0.32$, p=0.005. N: 99 dyads, 20 flies. (**I**) As in (**H**) for two groups of 10 virgin female Canton-S forming 45 dyads each according to scheme 3 and tested on successive days, $r(78) = 0.24$, p=0.034. (**J**) Visualization of a Social Interaction Network (SIN) of 10 Canton-S (45 dyads) in arenas with open-clear barriers. Connections denote dyads exhibiting absolute interactivity index values greater than the average of the absolute values of the 1st and 4th quartiles (threshold: 0.031). Threshold was identical for day 2. Colors and numbers indicate fly identity on both days.
DOI: https://doi.org/10.7554/eLife.37166.025

The following figure supplements are available for figure 5:

**Figure supplement 1.** Interactivity index controls.
DOI: https://doi.org/10.7554/eLife.37166.026

**Figure supplement 2.** Social interactions and their persistence as measured by the coincidence index metric.
DOI: https://doi.org/10.7554/eLife.37166.027

differences in dyadic interactions. This is consistent with reports that visual cues can mediate social modulation of behavior (*Kim et al., 2012*). Consistently, anosmic *Orco* mutant flies (*Vosshall and Hansson, 2011*) showed a significant 65% reduced mean absolute interactivity index, even with open-clear barriers (p=0.00048 by t-test). Blind *NorpA* mutant flies (*Kim et al., 1995*) did not differ significantly from wild type flies in mean absolute interactivity index (p=0.26 by t-test). Flies with mutations in the *white* gene ($w^{1118}$) have reduced visual acuity (*Markow and Scavarda, 1977*) but showed a 50% increase in absolute interactivity index which was statistically significant (p=0.013 by *t*-test), suggesting that social partner visual recognition does not require fine acuity (*Justice et al., 2012*). The increased inter-dyad variability seen in $w^{1118}$ animals may be consistent with increased inter-individual variability in phototactic preference exhibited by this genotype (*Kain et al., 2012*). In a control experiment, $w^{1118}$ flies separated by solid-black barriers had interactivity indices indistinguishable from Canton-S flies separated by solid-black barriers (p=0.48 by t-test).

Lastly, we set out to determine if social behavior is stable over long periods of time by measuring dyadic interactivity indices across days. Using our high-throughput dyad scheme, we measured interactivity indices from 91 dyads on each of 6 consecutive days. Interactivity indices across Canton-S dyads were statistically significantly correlated ($0.31 < r < 0.38$, $0.0001 < p < 0.04$) between day one vs. day 2, day 2 vs. day 3 and day 3 vs. day 4 (*Figure 5G*), but not days 4 vs. 5 or 5 vs. 6. Thus, dyad-specific affiliative behavior appears to be stable over days-long timescales, though only for the first days of experiments. In control experiments, Canton-S flies with solid-black barriers and *NorpA* and *Orco* mutant flies exhibited no significant correlation in dyad interactivity indices across days. Significant correlation in interactivity index across dyads was observed in our two SIN dyad schemes as well (*Figure 5H–J*) suggesting that SIN persistence can be measured in a variety of experimental formats. Other metrics of social interaction yielded qualitatively similar results to the interactivity index (*Figure 5—figure supplement 2*; Supplementary methods).

## Discussion

We developed MAPLE, a modular, automated platform for large-scale experiments, to expand automated experimental capabilities for lab model organisms (*Figure 1*). By design, MAPLE is versatile, scalable, and relatively inexpensive, with all components of the core fly-handling

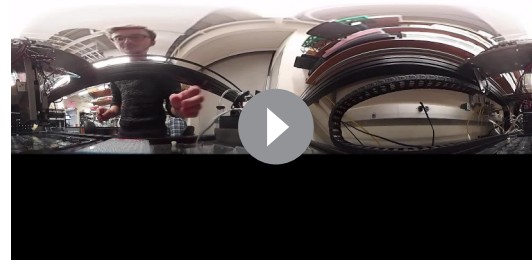

**Video 11.** Motion VR view from the MAPLE workspace floor. Time-lapse view from a wide-angle camera mounted on MAPLE's experimental workspace plate during a procedure to move flies from a 96-well plate into social arenas. Moments when MAPLE flashes multiple times successively in the same position reflect an algorithm to vary the exposure time for image acquisition to detect the opening of a multi-position loading port. This video is available on YouTube at https://youtu.be/F9KdnkGfkhI. The viewing angle can be adjusted during playback in the YouTube viewer.
DOI: https://doi.org/10.7554/eLife.37166.028

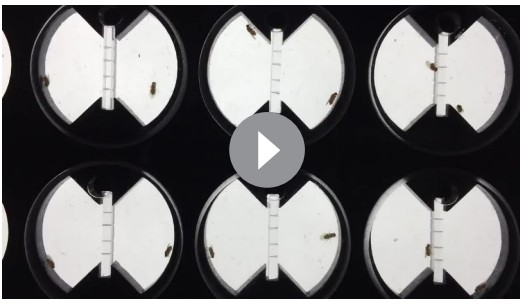

**Video 12.** Fly behavior in social arena assay. Six social arenas are seen with a fly loaded into each of the 12 semicircular compartments. These compartments are separated by open-clear barriers which are made of clear acrylic with horizontal channels allowing flies to both see and smell each other.

DOI: https://doi.org/10.7554/eLife.37166.029

robot costing roughly $3500. We displayed the versatility of this system by adapting the MAPLE platform to detect, place, and streak out individual *S. cerevisiae* colonies (*Figure 1B*), autonomously record *C. elegans* behavior (*Figure 1C*) and *P. polycephalum* movement (*Figure 1D*), as well as automatically monitor the behavior of multiple bumblebee colonies over long periods (*Figure 1F,G*). We have made the design and code for MAPLE open access in the hope that it, or descendant approaches, will be adopted by our field.

After demonstrating that MAPLE did not discernibly alter the baseline behavior of animals compared to manual handling (*Figure 2*), we developed a procedure by which MAPLE could help conduct a repetitive lab chore that nearly all *Drosophila* biologists are familiar with — collecting virgin females for genetic crosses. Using a Fly Dispenser, MAPLE collected individual animals as they eclosed from the pupal case and kept them in isolation, ensuring their virginity (*Figure 4*). To arrive at a final collection of virgin females, these singly-housed individuals were sexed by hand. Even with this manual step, virgins were procured at a much higher (~3 x) rate with MAPLE's assistance than without it (measured in terms of human labor time per virgin female).

MAPLE's capacity to handle individual flies in combinatorially complex experimental designs (*Figure 3*), allowed us to demonstrate that dyadic fly social behavior relies on both visual and olfactory cues. Dependence of social behavior on visual (*Mery et al., 2009*; *Simon et al., 2012*) and olfactory cues (*Schneider et al., 2012*; *Billeter and Levine, 2015*) is known from experiments observing many flies simultaneously. We further found that individual differences in dyadic social behavior — as well as derived social interaction network topography — remain stable over days (*Figure 5*). Stability of social networks on short timescales (*Schneider et al., 2012*) is known, but long-term stability of social networks is a novel finding. This illustrates that MAPLE can (1) replicate known results without introducing significant behavioral confounds and (2) extend our knowledge by conducting complex, long-term experiments that are challenging for human experimentalists.

Technologies that automate experiments will particularly facilitate the phenotyping of individual animals, an approach that reveals underappreciated intragenotypic variability. Our group has shown that individual flies generally exhibit very different behaviors from one another even if they are reared in the same environment and have the same genotype, and that these differences persist across days (*Kain et al., 2012*; *Buchanan et al., 2015*; *Ayroles et al., 2015*; *Kain et al., 2015*; *Todd et al., 2017*). MAPLE allowed us to show that this extends to dyad-specific social interactions in flies (*Figure 5H–J*). To our knowledge, this is the first demonstration that dyad-specific social interactions are persistent on days-long timescales in flies.

As the behavioral assay used here relies on physically separating individual flies, it is limited to examining visual and olfactory aspects of social interactions. To describe persistence of dyad-specific social interactions in flies more completely, further investigation including somatosensory and gustatory information exchange is necessary (*Krstic et al., 2009*; *Schneider et al., 2012*; *Ramdya et al., 2015*). Moreover, this assay presents flies with a choice to interact with a particular partner, or not. Assays with groups of flies, where the alternative to interacting with a particular fly includes interacting with potentially many other flies, may produce different results (*Ramdya et al., 2017*), although interaction networks in a group context are at least partially stable over the course of experiments (*Schneider et al., 2012*).

MAPLE is designed to be a general purpose instrument, therefore it has some disadvantages compared to purpose-built devices that perform a single task. For example, while MAPLE's camera can be used to acquire time-lapse movies of yeast colony growth, *Physarum* plasmodium movement, or *C. elegans* locomotion, purpose-built worm imaging systems can provide comparable data with higher throughput (*Churgin et al., 2017*) or higher resolution (*Stern et al., 2017*). Likewise,

purpose-built yeast colony picking robots (e.g. BioMatrix, S and P Robotics) achieve higher performance on specific tasks like transferring colonies, although typically at a higher price. MAPLE's open, modular design is meant to facilitate customization and prototyping, so it may be a good platform for the development of new organism-handling hardware, for example, for the dissection and transfer of *Physarum* syncytium or individual worms. These and other experimental applications might rely on liquid-handling. MAPLE currently has no liquid-handling capabilities, but there are open source liquid-handling robotic systems such as Opentrons (http://opentrons.com) which could be integrated with MAPLE. Specifically, liquid-handling effectors could be installed on MAPLE z-axes, replacing some or all of the current effectors.

In its current configuration, MAPLE has some drawbacks. Its effectors move slightly slower than the hands of a trained experimentalist, so its advantages derive from it not growing tired or losing focus. MAPLE does not execute every action with 100% success. For example, attempts to remove an awake fly from a behavioral arena, using vacuum through a loading port, succeed approximately 90% of the time. Given this error rate, the experimental procedures that include removing flies from arenas use the camera to inspect the arena after each attempt to confirm if the fly has been successfully removed. Similarly, our procedure to transfer yeast colonies used the collision-detector to confirm that the object manipulator had successfully removed petri dish lids. These checks reduce the efficiency of MAPLE-conducted experiments (particularly compared to manual experiments), but mean that the operator can walk away from the device confident that MAPLE will eventually get it right. Given its redundancies and fail-safes, MAPLE can work autonomously for long periods (e.g. 12 hr for the *Physarum* experiment) without getting stuck.

A potential advantage of using MAPLE for fly experiments specifically is the option to avoid anaesthesia. The Fly Dispenser releases awake animals into the MAPLE system, and these can then be moved between the FlyPlate and behavioral arenas using the multi-position loading ports we developed. Avoiding anesthesia has multiple benefits, including reducing the distortion of behavioral (*Bartholomew et al., 2015*) and physiological measurements (*Colinet and Renault, 2012*), which, depending on the form of anaesthesia, can last for hours or days (*MacMillan et al., 2017*). Even seemingly benign manual aspiration can disrupt the expression of sensitive phenotypes (*Trannoy et al., 2015*). Automated, anesthesia-free animal-handling thus has the potential to standardize handling effects.

MAPLE's modularity means that the platform's versatility and capabilities can be expanded in the future. As examples, we are developing fly sink modules for culturing thousands of animals in the style of population cages, facilitating selection experiments and experimental evolution. We are developing a high-resolution imaging source/sink module into which flies could be deposited, imaged in dorsal and ventral views, and released back into experimental workflows. This device is inspired by the Fly Catwalk system (*Medici et al., 2015*) but can be loaded with a fly on demand, rather than relying on flies to enter the imaging chamber on their own. The images from this module will allow machine-learning-based classification of morphological phenotypes, like sex and eye color, as well as quantification of fluorescent proteins; key genetic markers in flies.

In addition to conducting tedious or complex experiments, MAPLE's integrated format affords the opportunity to feed back the results of phenotypic assays into fly handling tasks. For example, MAPLE could be used to perform artificial selection, identifying individuals with a specific behavior or phenotype, and placing males and females together in wells of a FlyPlate, vials, or population cage. Likewise, multigeneration genetic crosses will be possible with our high resolution imaging module. Our current behavioral phenotyping relies on simple motion tracking, but behavioral assays employing thermogenetic or optogenetic stimulation, or sophisticated stimulus control could be implemented through custom modules. Unsupervised learning algorithms have been used to automatically phenotype flies (*Berman et al., 2014*; *Todd et al., 2017*), and with the capacity to store and access large numbers of flies individually, MAPLE could identify and isolate outliers within a population. MAPLE is compatible with such modern, automated approaches to fly experimentation, and brings automated animal-handling one step closer to the potential achieved by liquid-handling robots for molecular research.

## Materials and methods

CAD files for MAPLE can be found at https://github.com/FlySorterLLC/MAPLEHardware (copy archived at https://github.com/elifesciences-publications/MAPLEHardware). Control software for MAPLE including scripts for the experiments described here can be found at https://github.com/FlySorterLLC/MAPLEControlSoftware (copy archived at https://github.com/elifesciences-publications/MAPLEControlSoftware). Raw data and analysis scripts can be found at http://dx.doi.org/10.5281/zenodo.1119131. These materials are also available at http://lab.debivort.org/MAPLE.

### MAPLE technical specifications

MAPLE's frame is a rectangular prism constructed from extruded aluminum struts (Misumi HFS-5 series in various sizes and lengths) and brackets (Misumi five series). The principal axes of the robot are Cartesian, that is to say linear and mutually orthogonal. The longest axis - designated the X axis - comprises two supported linear rails (IGUS Drylin AWUM-12), each with two housed bearings (IGUS drylin OJUM-6–12). The total of four housed bearings support a single, wide linear rail (IGUS WS-10–120) that is the Y axis. A carriage made of two aluminum plates sandwiching four more bearing blocks (WJ200UM-01–10) slides along the Y axis. Suspended from this carriage is an assembly that houses the three independent Z axis slides (IGUS SLN-D740679-2). We deliberately chose sliding bearings (as opposed to ball bearing slides) for the three axes to avoid noise and vibrations that might confound behavioral experiments.

The first of the three end-effectors is an object manipulator. Made from an off-the-shelf vacuum cup connected to an air manifold, it can pick up and deposit lids or other small components from the workspace. A high-resolution digital camera and lens (The Imaging Source DFM 72BUC02-ML and TBL 9.6-2 C 3MP) is mounted to the second Z axis. Moving this Z carriage up and down focuses the camera. The third motorized Z slide holds a custom aspirator to move flies. Several short lengths of small diameter tubing (sections of needle tips, McMaster 75165A553 and 75165A682) are fixed approximately 4 mm inside a blunt, Luer-lock needle (McMaster 6710A61), forming a barrier to flies but allowing air to flow. A custom-molded silicone rubber boot surrounds the fly vacuum tip for the object manipulator, forming a better seal against apertures in modules and improving fly handling (*Figure 1—figure supplement 3*).

Stepper motors (1.5 A NEMA 17 60 mm bipolar stepper motor, MNEMA17-60 from RobotDigg.com) mounted to the main frame drive two 6 mm-wide GT-2 belts arranged in a CoreXY configuration (*Moyer, 2012*), and these belts drive the motion in the X and Y directions. Each Z axis slide is driven by its own stepper motor, integral to the slide assembly. The maximum speed for the X and Y axes is 200 mm/s. The Z axes top out at 83.3 mm/s. Limit switches (Omron SS-5) are mounted to the frame for each axis, providing repeatable end stops for homing.

Motion control, as well as the control of auxiliary devices such as the solenoid valves and the LED illumination, is handled by a Smoothieboard v1 PCBA, running custom Smoothieware firmware (included in our Github repository). G-code commands are sent from a PC connected via USB, interpreted on the Smoothieboard, and translated into electrical signals sent to each stepper motor. Scripts containing each experimental protocol, along with a common set of frequently used subroutines, were written in Python 2.7 (or Matlab 2016a for BEECH) and executed on PCs running Windows 7.

### Software

Every module has an associated Python 2.7 module class file (all MAPLE control software, including module class files, are available at https://github.com/FlySorterLLC/MAPLEControlSoftware; copy archived at https://github.com/elifesciences-publications/MAPLEControlSoftware). These files provide the 3D coordinates of key points on each module, including the sites of any ports or adapters through which flies are conveyed, as well as the z-clearance above the module needed so that the end-effectors do not collide with the modules in the workspace. These classes can be instantiated in Python files that represent workspace configurations (*Examples/ExampleWorkspace1.py*). Beyond the module class files, there is a master MAPLE class file (*robotutil.py*), which 1) establishes the communications connections between the experiment-coordinating computer and the motion control card and camera in MAPLE and 2) contains functions for all low-level robot operations, like returning

to the 0,0,0 home position, moving to an arbitrary position, opening or closing the solenoid valves that control the vacuum flow in the end effectors, or acquiring a photo from the end-effector camera. There is also a file (*commonFlyTasks.py*) that contains subroutines for common usage tasks, like the combination of end effector movements, vacuum and air valve engagement, and Fly Dispenser serial commands required to retrieve a fly from the Fly Dispenser adapter on the workspace. Files of an additional type, experimental scripts, implement the actual experimental procedures (e.g., *Figure 3—figure supplement 3*; *Video 4*). Each of these scripts load the class files for the modules used in their respective experiments, and procedurally calls the low- and mid-level functions of robotutil.py to implement each procedure. See *Figure 3—figure supplement 2* for a schematic of the software architecture. Lastly, to increase the convenience of using MAPLE, we implemented a remote control system in which representations of the current status of experiments (e.g. text reports and digital images) are posted to a dedicated email account. This account can also receive MAPLE commands by email to remotely trigger experimental procedures.

## Workspace

The workspace refers to a 100 cm x 28.2 cm x 7.5 cm volume that can be accessed by all end-effectors. As specified, the workspace is attached to the same frame as the axis carriages, but could be mounted separately to isolate experimental modules from the vibrations of MAPLE's motors. The bottom of this volume is a clear acrylic floor with 5 mm holes organized in a grid (10 cm apart) and cable pass-throughs for easy organization of individual modules. The 5 mm holes can be used to affix an interchangeable acrylic plate (a 'workplate') to the acrylic workspace floor using nylon thumb screws. Workplates have locating brackets which define the positions of modules in a particular experimental configuration. Thus, every experimental script is associated with a physical workplate that locates the modules used in its experiment.

## Modules

Modules were made by a variety of fabrication techniques, including laser-cutting of acrylic. Vector outlines of module components were made in Autodesk Inventor or Adobe Illustrator and laser cut by way of CorelDraw. Acrylic components were joined together using Plastruct plastic weld. Phenotyping modules used to investigate affiliative behavior and locomotor handedness, that is, social arena and y-maze arrays, measured 30 cm x 30 cm (*Figure 3*) and were fabricated from sheet acrylic using a laser cutter. Fly source modules were custom made ($CO_2$ pad) or purchased from FlySorter LLC. Fly Dispenser dimensions are 22 cm x 15 cm, $CO_2$ pad dimensions are 25 cm x 15 cm, and FlyPlate dimensions are 16 cm x 10 cm.

## Fly dispenser

The Fly Dispenser isolates and outputs individual flies from an attached vial. Repeated knocking motion (which mimics the tapping gesture that people use to knock flies down in a vial) causes flies to fall from the vial into a funnel. At the bottom of the funnel, a pair of motorized, soft foam wheels acts as a valve, and a photo interrupter detects when one fly has passed by. The wheels are stopped, preventing other flies from passing through the valve, and an air pump transports the isolated fly out of a tube. The dispensing process can be remotely triggered and monitored by Python scripts via a USB serial interface.

## Dispenser adaptor

MAPLE interfaces with the Fly Dispenser through the dispenser adaptor logistics module. The dispenser adaptor is a 4 cm x 1.5 cm 3D-printed ABS block with two 5 mm diameter plastic Luer lock tube sockets attached on opposite sides. The Dispenser handpiece connects to the bottom side of the dispenser adaptor, while the MAPLE fly manipulator end effector aligns to the opening on the top of the adapter.

## FlyPlate

The FlyPlate is a modified 96-well plate positioned on a food tray. Each well in the FlyPlate follows the 96-well plate standard for bottomless wells (7 mm in diameter and 10.9 mm deep). Wells have a stainless steel mesh floor that allows feeding but prevents escape. The plate lid has x-shaped laser-

cut openings over each well, cut into a flexible nylon mesh, that allow MAPLE's individual fly manipulator (or a handheld aspirator) to penetrate to remove or deposit flies. The openings close back up once the aspirator/manipulator tip has been removed, keeping flies securely housed. Flies had free access to standard cornmeal diet on the food tray placed below. Food was replaced every 2 days to maintain adequate moisture and freshness and remove eggs and first instar larvae.

### Morgue
The morgue fly sink module is a 10 cm diameter x 5 cm deep laser-cut acrylic cylinder covered by a detachable lid that allows quick disposal of its contents and is equipped with an nylon mesh adapter (in the style of the well coverings of the FlyPlate) that allows MAPLE to deposit flies into soapy water or ethanol that traps and euthanizes them.

### Fly food vial
The fly food vial is a standard 2.6 cm x 10 cm vial equipped with a detachable lid that facilitates MAPLE fly depositing. A standard fly culture media vial can be placed into the fly food vial.

### Behavior arena arrays
Behavior phenotyping modules can receive flies in one of two different ways, depending on whether the flies are anesthetized or not. In a traditional experimental style (*Ayroles et al., 2015*; *Buchanan et al., 2015*), MAPLE can pick up anesthetized flies from the $CO_2$ pad module with the organism manipulator, pick up the plastic lid covering a behavioral arena with the object manipulator, drop the fly in the arena, and replace the lid. In a MAPLE-optimized experimental style, awake flies are retrieved from the Fly Dispenser or FlyPlate and then transferred directly into the behavioral arena by sliding a multi-position loading port, a slidable clear lid with a 3.5 mm diameter opening through which flies can be deposited and removed, into place above the arena, dropping the fly, and then sliding the port so it is inaccessible to the fly (*Figure 3—figure supplement 1* and examples below).

### Social arena array
Arenas used for affiliative behavior (*Figure 5*) and circling bias experiments (*Figure 2—figure supplement 1*) are circular in shape with a 30 mm diameter and a height of 3 mm. Arenas are covered by a multi-position loading port. Two equal-sized semicircular compartments are formed by a 1.5-mm-thick interaction barrier. Interaction barriers refer to individually laser-cut blocks that can be placed into corresponding openings in the middle of the circular arena, allowing separation of flies into individual compartments. Barriers were laser-cut from either clear or black acrylic and were designed to be either solid or open. Solid barriers were flat on the bottom. Open barriers had 4 ~ 0.25 mm horizontal channels connecting the two compartments (*Figure 5B*; *Video 12*). Open barriers presumably facilitate the exchange of odor cues between compartments. Barriers could be made of clear acrylic, facilitating visual cues, or black acrylic. A social arena array comprises 81 circular arenas, with 162 semicircular arenas in total.

### Y-maze array
Locomotor handedness was assessed using y-maze arenas (*Buchanan et al., 2015*). Individual arms of the symmetrical Y-shaped mazes are 15.5 mm long and 120° apart. Arm ends are circular (5.2 mm) in shape, making it easier for flies to turn around and permitting loading and unloading flies via multi-position loading ports (*Figures 1*). Arenas are covered with identical lids as those in the social arena array. A y-maze array comprises 81 y-maze arenas arranged equidistantly in a nine-by-nine grid. All parts described were manufactured from either clear or black acrylic and cut into shape using a laser cutter.

### Yeast
Yeast expressing GFP in mitochondria (genotype can1-100 leu2-3,112 his3-11,15 ura3-1 BUD4-S288C RAD5 TRP, mdh1::KAN) were streaked from frozen stocks onto complete supplement mixture media lacking adenine (CME-Ade) and cultured at 30℃ for 1–2 days. Cells of this genotype express GFP in mitochondria, and are thus fluorescent under the dissecting scope. MAPLE touched single

colonies on the CME-Ade plate and streaked this material onto empty yeast extract-peptone-dextrose (YPD) plates. These target plates were allowed to incubate overnight at 30°C prior to imaging.

## Caenorhabditis elegans

We imaged N2 worms of mixed sexes and ages on standard growth media plates consisting of nematode growth medium in 1.7% agar with an OP50 *E. coli* lawn. Locomotion was recorded at 1 Hz at room temperature (21°C).

## Physarum

We inoculated petri dishes containing 2% agar in water with ~3 mm diameter excised pieces of an oatmeal-fed, actively growing *Physarum* plasmodium. These were allowed to recover from excision and plating for ~12 hr at 21°C prior to the collection of the 12 hr 0.017 Hz time-lapse movie in MAPLE, which was also conducted at 21°C.

## Fly lines

All experiments were performed using Canton-S (wild type), *Orco*, *NorpA*, or $w^{1118}$ lines. Mutant lines were homozygous. We raised flies on CalTech formula cornmeal mediaunder 12 hr/12 hr light and dark cycle in an incubator at 25°C and 70% humidity. Flies were anesthetized using carbon dioxide ($CO_2$) and housed in vials of 15 to 20 flies, unless otherwise specified. Five days post-eclosion, flies were aspirated into individual wells in the FlyPlate using $CO_2$ and used for experimentation after at least 2 hr of recovery.

## General fly experimental procedures

All experiments were conducted between 9AM and 9PM (ZT0-ZT12). Flies were loaded into individual arenas by MAPLE; arenas that remained empty after two iterations were loaded manually using an aspirator. Flies were later removed from their arenas in an identical fashion. FlyPlates, social arenas, and y-maze arrays were filled with 96, 162, and 81 flies, respectively. Flies were assayed using diffused white LED backlighting (*Buchanan et al., 2015*) in a temperature (23°C) and humidity (41%) controlled behavioral observation room. Fly movement was tracked for 1 hr. Fly tracks were analyzed using a custom MATLAB script. For longitudinal assaying, flies were moved back into the FlyPlate after phenotyping and allowed to feed and rest overnight or for 1 hr at minimum.

## Behavior measurement

Fly movement was tracked at 29.9 fps using a custom real-time MATLAB script interfacing with a Firefly MV FMUV-13S2C USB-camera. Tracks were analyzed in MATLAB. In total, 3% of the data were discarded because flies were immobile as determined by mean speed thresholds.

## Statistics

Unless otherwise specified, all confidence intervals were computed by bootstrapping the data associated with individual flies or individual fly dyads (in the case of social interaction measurements) 1000 times, using custom MATLAB scripts. One standard deviation of the bootstrap estimates was our estimate of the standard error of the estimate. P-values reported were adjusted for multiple comparisons using the Bonferroni correction where applicable, and asterisks reflect post-correction significance.

## Specific experimental procedures

### Activity and speed MAPLE handling control experiments

Flies were loaded into a FlyPlate in accordance with general experimental procedures. MAPLE removed every second fly from its individual well and released it back after a 1 s delay 10 times in a row (48 flies total MAPLE-handled). This procedure was repeated three times so that every second fly was handled 30 times in total after 1 hr (*Figure 2A–D*). Flies were then monitored according to general experimental procedures. The preceding steps constituted one handling session. There were five handling sessions lasting 10 hr in total.

## Manual vs MAPLE-handling control experiments

Flies were loaded into the y-maze array according to general experimental procedures (*Figure 2E–H*). MAPLE loaded a random arena compartment (81 flies total) of each arena in the social arena tray to prevent dyad-neighbors influencing circling behavior. Flies were observed as described in general experimental procedures. Flies were discarded into morgue fly sink module after phenotyping.

## Virgin-picking procedure

Ten male and 20 female CS between 5 and 7 days post-eclosion are placed in custom dispenser-type vials. These custom virgin-picking vials are 10 cm long x 4 cm in diameter open cylinders with a bottom that attaches by press-fit. Standard cornmeal diet was poured in the bottom portion and allowed to cool down prior to fly introduction. Flies were anaesthetized using $CO_2$ and placed in the vial. After 2 days of egg-laying in the incubator (25℃), parental flies were discarded and the vial was placed back in the incubator. After 9 days, the bottom portion of the vial containing the food was removed from the container. The food was discarded and the container washed and reattached to the vial. The vial now only contained animals that pupated on the sides of the open cylindrical portion of the vial (*Video 10*).

The vial was then placed into the Fly Dispenser and MAPLE's virgin-picking subroutine was engaged. Every 30 min, the Fly Dispenser attempted to dispense any eclosed flies while MAPLE aligns its fly manipulator end effector to the fly dispenser adapter (*Figure 3—figure supplement 3A*). If a fly was successfully dispensed, MAPLE deposited it into a FlyPlate in the workspace. If at any point no fly is dispensed, MAPLE and the Fly Dispenser paused for a 30-min waiting period before a new dispensing attempt was made. Over 3 days, MAPLE continued to load newly eclosed flies into individual FlyPlate wells until pupa were exhausted (*Video 10*). FlyPlate wells containing multiple flies were manually emptied with an aspirator. The remaining flies were manually anesthetized using $CO_2$ and their sex assessed under a dissecting microscope. After sexing, flies were returned to their individual wells. The FlyPlate fly source module including food tray was removed from the workspace and placed inside a sealed plastic container. After 10 days, the food tray was examined for larvae, eggs, and newly hatched flies. When none were found, the single flies MAPLE placed into individual FlyPlate wells were considered virgins.

## Virgin-picking MAPLE/manual comparison

To allow a fair head-to-head comparison between manual virgin collection and MAPLE-assisted virgin collection, our approach was to start and end both procedures in identical circumstances (bottles of parental flies, and vials containing virgin progeny females, respectively). Two standard bottles containing 10 male and 20 female CS flies 5–7 days post-eclosion were allowed to mate and lay eggs for 3 days. After 3 days, flies were removed and egg- and larvae-containing agar was transferred with a spatula among five standard vials and one custom dispenser-type vial. The amount of virgin females automatically picked by MAPLE was assessed according to the virgin-picking procedure described above. Total time required was computed as the total time required for every step of the process, including preparation, manual sexing, and cleanup. Virgin female count, virgin female ratio, and time required for manual virging was assessed by twice daily (9AM and 9PM) manual virgin-picking from five virgin-producing standard culture vials.

## Social interaction paradigm validation

Social arenas were prepared by manually inserting the appropriate interaction barrier type into the arena array prior to introducing flies. Flies were loaded into social arena arrays according to general experimental procedures. To minimize behavioral confounds caused by disparate loading times, one social arena compartment (i.e. all the left compartments) was filled first. This ensured that flies in social arenas loaded earlier were allowed minimal additional time to familiarize with or habituate to their dyad-neighbor. Phenotyping arrays were moved into behavior-recording boxes according to general experimental procedures. After assaying, flies were manually removed from trays and discarded.

### Social behavior day-to-day persistence (Scheme 1)

Flies were loaded into social arena arrays, assayed, and deposited into FlyPlates after phenotyping according to general experimental procedures. Dyads were randomly determined on the first day. Flies were phenotyped once per day. Fly identity and dyad composition was maintained throughout 6 assaying days. Compartments and arenas were loaded in a randomized fashion each day. On the 7th day, Canton-S flies in the open-clear interaction barrier condition were randomly placed in social arena compartments to form physically shuffled dyads to complement computational shuffling for resampling statistics.

### Social behavior persistence (Scheme 2)

Flies were loaded into social arena arrays, assayed, and deposited into FlyPlates after phenotyping according to general experimental procedures. Each fly was randomly assigned to be part of 10 dyads on the first day. Flies were assayed six times per day for 7 days. Fly identity was maintained throughout the experimental duration by storing flies individually. In total, each dyad was phenotyped four times. We averaged behavioral measures across the first and last two phenotyping sessions.

### Social interaction network (SIN) persistence (Scheme 3)

Flies were born in the Fly Dispenser and deposited into FlyPlates according to the virgin-picking procedure. Two groups of 10 flies each were assayed and deposited back into FlyPlates after phenotyping according to general experimental procedures. Flies were assayed six times per day for 3 days to exhaust each possible dyad combination twice.

## Social interaction analyses

### Interactivity index

A dyad's interactivity index was defined as the correlation of dyad-neighbors' distances to the interaction barrier over the entire experimental duration (*Figure 5B–D*). Distance was defined as the euclidean distance between a fly's centroid and the closest side of the interaction barrier.

### Coincidental approaches

We defined a coincidental approach as an interval in which both flies in a dyad were located within one body-length (3 mm) of the interaction barrier on the same frame. This definition of social interaction yielded qualitatively similar results to the interactivity index (*Figure 5—figure supplement 2*). Distance to the barrier was defined as the euclidean distance between a fly's centroid and the nearest side of the barrier. A coincidental approach was scored as a single event irrespective of its duration. For each subsequent coincidental approach to be valid, at least one fly was required to leave and re-enter the 3 mm zone. Coincidental approaches were normalized for dyad mean speed over the experimental duration. Dyad mean speed was the grand mean of both dyad-neighbors' mean speed.

### SIN connection threshold

In the graph representation of social interactions, edges between flies in the network were retained if the absolute value of their interactivity index was greater than the mean of the first and third quartiles of all dyads' absolute interactivity indices. The same threshold was applied to both repetitions of the SIN measurement (*Figure 5J*).

## Acknowledgements

We are grateful to Jim Morris for assistance developing the Smoothieware firmware, Matt Zucker for ongoing software and hardware advice, Chris Stokes for assistance with early prototypes, Yong-Li Dich for help with hardware development, and Kyobi Kakaria for consultations, fabrication assistance, and manuscript feedback. We thank the manuscript reviewers for constructive feedback. Cara Weisman helpfully guided us on the yeast protocol and supplied us with yeast and media plates. Vladislav Susoy helpfully guided us on the *C. elegans* protocol and provided us with worms and media. Jess Kanwal, Katrin Vogt, and Armin Bahl provided helpful manuscript feedback. Claire Guérin

helped with bee colony box design and BEECH assembly. BdB was supported by a Sloan Research Fellowship, a Klingenstein-Simons Fellowship Award, and the National Science Foundation under grant no. IOS-1557913. JDC was supported by a Winslow Foundation research grant. ABK was supported by a James S McDonnell Foundation Postdoctoral Fellowship Award in Studying Complex Systems. DZ was supported by the NIH under Award Number R43OD023302. The content is solely the responsibility of the authors and does not necessarily represent the official views of the National Institutes of Health.

## Additional information

### Competing interests

Dave Zucker: Dave Zucker is the founder and manager of FlySorter, LLC, which was originally included as his primary and sole affiliation. Benjamin L de Bivort: Benjamin de Bivort is on the scientific advisory board of FlySorter, LLC. The other authors declare that no competing interests exist.

### Funding

| Funder | Author |
| --- | --- |
| Alfred P. Sloan Foundation | Benjamin L de Bivort |
| Esther A. and Joseph Klingen-stein Fund | Benjamin L de Bivort |
| National Science Foundation | Benjamin L de Bivort |
| Winslow Foundation | James D Crall |
| James S. McDonnell Foundation | Albert B Kao |
| National Institutes of Health | Dave Zucker |

The funders had no role in study design, data collection and interpretation, or the decision to submit the work for publication.

### Author contributions

Tom Alisch, Resources, Data curation, Software, Formal analysis, Validation, Investigation, Visualization, Methodology, Writing—original draft, Writing—review and editing; James D Crall, Albert B Kao, Resources, Software, Investigation, Methodology, Writing—original draft, Writing—review and editing; Dave Zucker, Conceptualization, Resources, Software, Supervision, Funding acquisition, Validation, Investigation, Methodology, Writing—original draft, Project administration, Writing—review and editing; Benjamin L de Bivort, Conceptualization, Software, Formal analysis, Supervision, Funding acquisition, Visualization, Methodology, Writing—original draft, Project administration, Writing—review and editing

### Author ORCIDs

Tom Alisch http://orcid.org/0000-0001-7884-7031
James D Crall http://orcid.org/0000-0002-8981-3782
Dave Zucker https://orcid.org/0000-0003-0259-6293
Benjamin L de Bivort http://orcid.org/0000-0001-6165-7696

### Decision letter and Author response

Decision letter https://doi.org/10.7554/eLife.37166.036
Author response https://doi.org/10.7554/eLife.37166.037

## Additional files

### Supplementary files

• Supplementary file 1. MAPLE bill of materials. Spreadsheet listing MAPLE's components, parts numbers, vendor links, and fabrication techniques.
DOI: https://doi.org/10.7554/eLife.37166.030

• Supplementary file 2. MAPLE Build Instructions. Document outlining the steps to assemble MAPLE from parts, including frame, rails, effectors, control electronics, wiring and plumbing. Some maintenance information is included, as well as software instructions to test MAPLE's basic functionality and communication protocols.
DOI: https://doi.org/10.7554/eLife.37166.031

• Transparent reporting form
DOI: https://doi.org/10.7554/eLife.37166.032

### Data availability

CAD files for MAPLE can be found at https://github.com/FlySorterLLC/MAPLEHardware (copy archived at https://github.com/elifesciences-publications/MAPLEHardware). Control software for MAPLE including scripts for the experiments described here can be found at https://github.com/FlySorterLLC/MAPLEControlSoftware (copy archived at https://github.com/elifesciences-publications/MAPLEControlSoftware). Raw data and analysis scripts can be found at http://dx.doi.org/10.5281/zenodo.1119131. These materials are also available at http://lab.debivort.org/MAPLE.

The following dataset was generated:

| Author(s) | Year | Dataset title | Dataset URL | Database and Identifier |
|---|---|---|---|---|
| Tom Alisch, James D Crall, Dave Zucker | 2017 | Data for: Alisch et al MAPLE Supplementary Materials | http://dx.doi.org/10.5281/zenodo.1119131 | Zenodo, 10.5281/zenodo.1119131 |

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
