## [Decision Letter]

Thank you for submitting your article "MAPLE (Modular Automated Platform for Large-scale Experiments), a robot for integrated animal-handling and phenotyping" for consideration by *eLife*. Your article has been reviewed by three peer reviewers, including Ronald L Calabrese as the Reviewing Editor and Reviewer #1, and the evaluation has been overseen by a Reviewing Editor and Eve Marder as the Senior Editor. The following individuals involved in review of your submission have agreed to reveal their identity: Giorgio F. Gilestro (Reviewer #3).

The reviewers have discussed the reviews with one another and the Reviewing Editor has drafted this decision to help you prepare a revised submission.

Summary:

In this Tools and Resources manuscript, the authors report on developing a multifunctional robot system for handling model organisms. Specifically, they developed a Modular Automated Platform for Largescale Experiments (MAPLE), an organism handling robot capable of conducting lab tasks and experiments, and then adapted it with different organism specific modules and used it to conduct common experimental procedures in *Saccharomyces cerevisiae, Caenorhabditis elegans, Physarum polycephalum, Drosophila melanogaster*, and *Bombus impatiens*. They focused on its applicability to *D. melanogaster*. They show in detail how it can be used to reduce the time commitment to virgin collection with high fidelity. They also use it to study dyadic social interactions. They replicate previous experimental results on the role of visual and olfactory cues and specificity of dyadic interaction and produce the novel result that dyadic interactions persist over multiple days. Thus, they demonstrate it's potential in the laboratory. They provide detailed methods and open source plans for producing a MAPLE, and they convincingly argue that the modular design will allow innovation and adaption to new organisms and experimental procedures. This should be a useful tool for many research labs.

Essential revisions:

The discussion requested by reviewer #2 of potential liquid handling is important and should be directly addressed.

There is a specific concern about the social network biology raised by reviewer #2 that should be addressed.

There is a specific concern raised by reviewer #2 for comparisons to manual processes for animals other than flies that should be addressed.

Both reviewer #2 and #3 want a more thorough discussion of the limitations of MAPLE and provide clear indications of how to proceed in their comments.

All reviewers agree that the manuscript could be made more concise, especially given the extensive supplementary material.

Reviewer #2:

- There is no comparison of MAPLE's capability to manual experimenters for any other organism making it difficult to determine if the initial feasibility demonstrations represent true advances in throughput with automation or automation alone. For example, measuring videos of multiple plates of *C. elegans* is a rather trivial advance, as switching plates on any standard behavior tracking system takes only a few seconds. A real advance for *C. elegans* would be the ability to pick (or liquid-transfer) or chunk populations of automatically maintained worms to new plates without damaging them for phenotypic analysis.

- Does MAPLE have liquid-handling capabilities? If not, can it be adapted to? It would be ideal if a drug treatment and/or the same robot that handles animals could apply liquid or gaseous stimuli. For example, MAPLE could administer repeated doses of chemicals to a single fly at precise times over long time scales to see how this alters dyad behavioral structure. While it is likely beyond the scope of the present work to build in liquid-handling capabilities to MAPLE it would be beneficial to discuss this limitation and potential ways this important capacity might be incorporated into the system.

- The authors present their social arena arrays as a solution to the problem of maintaining object ID during population-based social interaction assays. While it does solve the problem of maintaining object IDs, this experiment now gives the flies a choice between a partner and a wall, not a partner or another fly. While novel data presented here are convincing that individual flies prefer certain other flies consistently, it is still not clear whether this individual preference would be consistent if in a population of other interacting flies/potential partners. This limitation and implications for understanding *Drosophila* social architecture should be discussed.

- There should be a succinct cross-species discussion and comparison of the extensive number of microfluidic devices designed for the same purpose as MAPLE. These systems allow for easy animal handling, automation and high-throughput phenotyping and are available for many of the organisms studied in this work. MAPLE offers several advantages of these systems that should be discussed. In addition, there should be a mention of caveats- of areas in which the system has limitations.

Reviewer #3:

The only weakness of this work is that it is not clearly stated what the limitations of the system are at this stage and how easily (if at all) they could be solved. The manuscript has a very general "optimistic" outlook and it is not clear for me as a reader whether MAPLE could work out of the box for my purposes, which may be quite different from the ones of the authors. The only reported weakness in the provided examples is the one shown in Figure 3. Perhaps the discussion could feature a more comprehensive analysis of strengths vs. weaknesses of the current system. Example of current weaknesses could be: can MAPLE set up crosses automatically beside collecting virgins? does it ever get stuck? what is the longest autonomous experiment in self-drive you performed? can multiple "phenotyping modules" operate at the same time?

Finally: I usually do not comment on writing style because I do enjoy reading manuscripts with different, personal touches. However, in this particular case, I think the manuscript would benefit quite a bit from being shortened. The Introduction, for instance, is quite repetitive and uses too many words to stress an important but relatively easy concept. Some of the results could probably be moved to supplementary or not being shown at all (for instance, Figure 3B does not really add much – it's enough to know what the error rate is. Knowing the position of the wrong vials in that particular trial is too much information).

This is ultimately a joint author/editorial decision, but my recommendation would be to shorten the manuscript quite a bit.

---

## [Author Response]

Summary:In this Tools and Resources manuscript, the authors report on developing a multifunctional robot system for handling model organisms. Specifically, they developed a Modular Automated Platform for Largescale Experiments (MAPLE), an organism handling robot capable of conducting lab tasks and experiments, and then adapted it with different organism specific modules and used it to conduct common experimental procedures in Saccharomyces cerevisiae, Caenorhabditis elegans, Physarum polycephalum, Drosophila melanogaster, and Bombus impatiens. They focused on its applicability to D. melanogaster. They show in detail how it can be used to reduce the time commitment to virgin collection with high fidelity. They also use it to study dyadic social interactions. They replicate previous experimental results on the role of visual and olfactory cues and specificity of dyadic interaction and produce the novel result that dyadic interactions persist over multiple days. Thus, they demonstrate it's potential in the laboratory. They provide detailed methods and open source plans for producing a MAPLE, and they convincingly argue that the modular design will allow innovation and adaption to new organisms and experimental procedures. This should be a useful tool for many research labs.

We thank the reviewers and editor for this consensus statement that recognizes the strengths of the manuscript and clearly summarizes the points for improvement.

Essential revisions:The discussion requested by reviewer #2 of potential liquid handling is important and should be directly addressed.There is a specific concern about the social network biology raised by reviewer #2 that should be addressed.There is a specific concern raised by reviewer #2 for comparisons to manual processes for animals other than flies that should be addressed.Both reviewer #2 and #3 want a more thorough discussion of the limitations of MAPLE and provide clear indications of how to proceed in their comments.All reviewers agree that the manuscript could be made more concise, especially given the extensive supplementary material.

We thank the reviewers and editor for this consensus statement that recognizes the strengths of the manuscript and clearly summarizes the points for improvement.

Reviewer #2:- There is no comparison of MAPLE's capability to manual experimenters for any other organism making it difficult to determine if the initial feasibility demonstrations represent true advances in throughput with automation or automation alone. For example, measuring videos of multiple plates of C. elegans is a rather trivial advance, as switching plates on any standard behavior tracking system takes only a few seconds. A real advance for C. elegans would be the ability to pick (or liquid-transfer) or chunk populations of automatically maintained worms to new plates without damaging them for phenotypic analysis.

In general, MAPLE does not move faster than a person, so head-to-head comparisons on single actions like loading a plate into a behavior assay will generally not come out in MAPLE’s favor. MAPLE wins on long, tedious procedures when human attention can flag (or complicated procedures where manual errors become likely). E.g., for the virgin-collecting, human labor time/virgin is much lower using MAPLE than traditional techniques, but robot time/virgin is higher, because MAPLE is slow on each step despite being autonomous. So, the reviewer is right that MAPLE isn’t faster than people at moving a camera between *C. elegans* plates (compared to manually loading plates into a traditional imaging rig). The benefit of MAPLE is in longitudinal imaging of those worms, e.g., MAPLE would not be challenged by imaging those plates through the night to collect circadian data. With modification, MAPLE could be adapted to provide any number of stimuli throughout the course of the (long duration) experiment, such as mechanical agitation, light or dark periods, repositioning of the dishes, the addition of chemicals, etc.

We recognize that picking worms would be a significant advance, and we have certainly not provided a tool to do this in MAPLE’s current configuration, which is most developed for fly work. However, MAPLE is meant to be flexible and modular and we think it represents as good a platform as any for the prototyping of a worm-picking technology. We now address this in the Discussion section.

- Does MAPLE have liquid-handling capabilities? If not, can it be adapted to? It would be ideal if a drug treatment and/or the same robot that handles animals could apply liquid or gaseous stimuli. For example, MAPLE could administer repeated doses of chemicals to a single fly at precise times over long time scales to see how this alters dyad behavioral structure. While it is likely beyond the scope of the present work to build in liquid-handling capabilities to MAPLE it would be beneficial to discuss this limitation and potential ways this important capacity might be incorporated into the system.

In its baseline configuration, MAPLE has no liquid-handling capabilities. However, we chose to publish MAPLE’s parts list and assembly guide in a fully open format so that modifications and upgrades like the addition of liquid-handling would be relatively easy. The z-carriages are basically flexible platforms on which any number of effectors, including liquid-handling units, could be attached. The tubing and wiring needed to operate liquid-handling effectors could be easily incorporated into MAPLE’s cable management system. Opentrons is an open-access automated liquid-handling system with hardware designs that could very likely be added to the MAPLE platform (or vice versa).

We now discuss liquid-handling in the Discussion section, and Opentrons specifically.

- The authors present their social arena arrays as a solution to the problem of maintaining object ID during population-based social interaction assays. While it does solve the problem of maintaining object IDs, this experiment now gives the flies a choice between a partner and a wall, not a partner or another fly. While novel data presented here are convincing that individual flies prefer certain other flies consistently, it is still not clear whether this individual preference would be consistent if in a population of other interacting flies/potential partners. This limitation and implications for understanding Drosophila social architecture should be discussed.

The reviewer is correct that our social paradigm as a choice between interaction with a specific fly or no fly at all, rather than one or more flies among a number of alternative partners. We have edited the description of the assay in the Results section to make this distinction clear, and we have edited the Discussion section to acknowledge this limitation.

- There should be a succinct cross-species discussion and comparison of the extensive number of microfluidic devices designed for the same purpose as MAPLE. These systems allow for easy animal handling, automation and high-throughput phenotyping and are available for many of the organisms studied in this work. MAPLE offers several advantages of these systems that should be discussed. In addition, there should be a mention of caveats- of areas in which the system has limitations.

We have added a succinct paragraph to the Discussion section that considers MAPLE’s capabilities with respect to other purpose-built model-system automation devices. In brief:

MAPLE is capable of basic operations involving yeast culture plates and the macroscopic transfer of yeast colonies. MAPLE can perform these operations in coordination with long-term imaging of the colonies. But MAPLE is not (in this incarnation) as fast as a human and is surely not as effective at conducting molecular biology protocols as purpose-built robots.

MAPLE’s *C. elegans* capabilities are comparable to its yeast capabilities. We are not currently aware of any automation technology that can pick and transfer individual worms.

MAPLE’s *Physarum* capabilities are again comparable. It is likely capable of transferring bulk pieces of *Physarum*syncytium. As there are no purpose-built devices for handling *Physarum*, this demonstrates potential uses of MAPLE for non-model species.

Reviewer #3:The only weakness of this work is that it is not clearly stated what the limitations of the system are at this stage and how easily (if at all) they could be solved. The manuscript has a very general "optimistic" outlook and it is not clear for me as a reader whether MAPLE could work out of the box for my purposes, which may be quite different from the ones of the authors. The only reported weakness in the provided examples is the one shown in Figure 3. Perhaps the discussion could feature a more comprehensive analysis of strengths vs. weaknesses of the current system. Example of current weaknesses could be: can MAPLE set up crosses automatically beside collecting virgins? does it ever get stuck? what is the longest autonomous experiment in self-drive you performed? can multiple "phenotyping modules" operate at the same time?

This is a valuable observation. We did not intend to present only the rosy picture of MAPLE. The new Discussion section paragraph suggested by reviewer #2, which compares MAPLE to purpose built devices for other model systems, goes some way to presenting the limitations of the system. In addition, we have answered the specific questions suggested by the reviewer in two new paragraphs of the Discussion section that focus on limitations. Here we also emphasize some of the limitations we had to overcome to make MAPLE perform the assays described. As an example, the vacuum to remove flies from behavioral arenas is not 100% effective on every attempt, so the experimental algorithm includes repetition and steps to use the camera to detect whether flies have been removed. This of course, reduces the efficiency of the automation, and we now make this explicit.

As to the reviewer’s last question, for the phenotyping modules we used here, we generally used MAPLE to load the device before transferring it to traditional imaging rigs for phenotype measurement. However, we left the space above and below MAPLE’s workspace largely open so that optical phenotyping could happen within the workspace itself. Or non-optical methods like capacitive or voltage-based sensing of fly position could be used to phenotype flies in the workspace without even relying on its open optical axes. The capability of running multiple phenotyping modules simultaneously within MAPLE’s workspace comes down to setting up the phenotyping in that setting.

Finally: I usually do not comment on writing style because I do enjoy reading manuscripts with different, personal touches. However, in this particular case, I think the manuscript would benefit quite a bit from being shortened. The introduction, for instance, is quite repetitive and uses too many words to stress an important but relatively easy concept.

We have edited the Introduction, Results section and Discussion section down so that they are more streamlined and hopefully more to the reviewer’s liking. We only trimmed the Introduction a bit, since it was already only three paragraphs. Unfortunately, given the other valid requests from the reviewers, the current version is slightly longer than the previous version. If the reviewers and editors are of a consensus that more should be trimmed we will look for ways to do so.

Some of the results could probably be moved to supplementary or not being shown at all (for instance, Figure 3B does not really add much – it's enough to know what the error rate is. Knowing the position of the wrong vials in that particular trial is too much information).

We think the reviewer is referring to Figure 4B, where we illustrated the results of a virgining procedure and the number of flies in each well? Under that assumption, we have moved this panel into a new supplementary figure. This may also help with the reviewer’s suggestion to trim the length of the manuscript.

This is ultimately a joint author/editorial decision, but my recommendation would be to shorten the manuscript quite a bit.